# Multivalency ensures persistence of a +TIP body at specialized microtubule ends

Sandro M. Meier [1,2,3,9], Ana-Maria Farcas[2,3,9], Anil Kumar[1,7,9], Mahdiye Ijavi[3,4], Robert T. Bill [2,8], Jörg Stelling [5], Eric R. Dufresne [3,4], Michel O. Steinmetz [1,6] ✉ & Yves Barral [2,3] ✉

Microtubule plus-end tracking proteins (+TIPs) control microtubule specialization and are as such essential for cell division and morphogenesis. Here we investigated interactions and functions of the budding yeast Kar9 network consisting of the core +TIP proteins Kar9 (functional homologue of APC, MACF and SLAIN), Bim1 (orthologous to EB1) and Bik1 (orthologous to CLIP-170). A multivalent web of redundant interactions links the three +TIPs together to form a '+TIP body' at the end of chosen microtubules. This body behaves as a liquid condensate that allows it to persist on both growing and shrinking microtubule ends, and to function as a mechanical coupling device between microtubules and actin cables. Our study identifies nanometre-scale condensates as effective cellular structures and underlines the power of dissecting the web of low-affinity interactions driving liquid–liquid phase separation in order to establish how condensation processes support cell function.

Microtubules play a central role in moving and positioning diverse cargos within eukaryotic cells and thus in controlling cellular architecture and physiology (reviewed in ref. [1]). To fulfil these functions, individual microtubules acquire specialized behaviours to carry out dedicated roles independently of each other. One large class of regulators contributing to microtubule specialization are microtubule plus-end tracking proteins (+TIPs), which can be broadly sorted into two groups. The first group comprises ubiquitously expressed and evolutionary conserved proteins that bind the plus-ends of most if not all microtubules (reviewed in refs. [2,3]). These include the end binding proteins (EBs, for example, EB1, EB2 and EB3 in metazoans), which directly bind microtubule plus-ends, and the cytoplasmic linker proteins (CLIPs, for example, CLIP-170 in metazoans), which bind microtubule plus-ends through EBs. These +TIPs typically track growing but not shrinking microtubule plus-ends. The second group of +TIPs, the recently described 'patterning +TIPs'[4], comprises protein families of diverse phylogenetic origins that typically associate with and specify the function of individual microtubules. Representatives include the microtubule–actin crosslinking factor (MACF/ACF7/Shot[5]), SLAIN motif-containing proteins 1 and 2 (ref. [6]) and adenomatous polyposis coli (APC (refs. [7,8])). These +TIPs bring together plus-ends of specialized microtubules with target cellular structures such as the actin cytoskeleton, anchorage sites at the plasma membrane, or chromosomes (reviewed in refs. [2,3]). However, what restricts these patterning +TIPs to dedicated microtubule ends is poorly understood.

The budding yeast *Saccharomyces cerevisiae* offers a powerful model for studying how microtubules become specialized. During mitosis, every cell comprises two spindle pole bodies (SPBs, functional metazoan centrosome equivalents): an old SPB, inherited from the previous mitosis, and a newly synthesized one (reviewed in ref. [9]). Each SPB nucleates a small aster composed of one to three cytoplasmic microtubules[10,11]. Interestingly, these cytoplasmic microtubules

[1]Laboratory of Biomolecular Research, Division of Biology and Chemistry, Paul Scherrer Institut, Villigen, Switzerland. [2]Department of Biology, Institute of Biochemistry, ETH Zürich, Zürich, Switzerland. [3]Bringing Materials to Life Initiative, ETH Zürich, Zürich, Switzerland. [4]Department of Materials, ETH Zürich, Zürich, Switzerland. [5]Department of Biosystems Science and Engineering and SIB Swiss Institute of Bioinformatics, ETH Zürich, Basel, Switzerland. [6]University of Basel, Biozentrum, Basel, Switzerland. [7]Present address: ImmunOs Therapeutics AG, Schlieren, Switzerland. [8]Present address: Department of Molecular Life Sciences, University of Zürich, Zürich, Switzerland. [9]These authors contributed equally: Sandro M. Meier, Ana-Maria Farcas, Anil Kumar. ✉e-mail: michel.steinmetz@psi.ch; yves.barral@bc.biol.ethz.ch

are specialized in an SPB-dependent manner. The old SPB forms long microtubules that reach to the bud cortex, the future daughter cell. In contrast, microtubules generated from the new SPB remain highly dynamic and short[12–15]. The cortical interactions of old SPB microtubules orient this SPB towards the bud[16,17] and align the intranuclear spindle with the future cell division axis, before anaphase onset. Spindle elongation during anaphase then distributes the genetic material equally between mother and bud[18–20].

Microtubule specialization at the old SPB relies on exclusive decoration of their plus-ends with the protein Kar9, a patterning +TIP. At the plus-end of one microtubule, which Kar9 binds through the EB-family member Bim1, Kar9 interacts with the CLIP-family member Bik1 and the type V myosin Myo2, an actin-directed motor protein[4,8,21–23]. Myo2 then pulls the microtubule plus-end along actin cables to drive its specific interaction with the bud cortex[24,25]. A unique feature allowing Kar9 to remain associated with a chosen microtubule throughout mitosis is its ability to track its plus-end throughout phases of both microtubule growth and shrinkage[19,20]. However, the mechanisms of this behaviour are unknown. In this Article, to address the fundamental issue of microtubule specialization, we investigated the mechanisms of Kar9 cohesion and persistence to essentially one dynamic microtubule plus-end in vivo through dissecting the interaction network that Kar9 establishes with its key partners Bim1 and Bik1.

## Results

### Kar9-network components undergo phase separation in vitro

To characterize the interactions of the Kar9-network core components with each other, we recombinantly produced and purified Kar9, Bim1 and Bik1, and monitored their interactions in vitro using a pelleting assay. Mixed together in stoichiometric amounts (10 µM each), the three proteins were soluble in a standard, crowding-agent-free and pH-neutral buffer containing 500 mM sodium chloride, but pelleted readily upon ultracentrifugation when the salt concentration was reduced to 200 mM (Fig. 1a). Phase contrast microscopy and fluorescence recovery after photobleaching (FRAP) experiments indicated that under low-salt condition the proteins phase separated into liquid-like droplets in which Kar9 and Bim1 diffuse slowly and Bik1 more rapidly (Fig. 1b, Extended Data Fig. 1 and Supplementary Movie 1). The mixtures spiked with low level of green fluorescent protein (GFP)-tagged versions of the proteins indicated that Bik1, Bim1 and Kar9 were enriched 8, 27 and 15 times, respectively, in these droplets (Extended Data Fig. 2a). Removal of Bim1 or Bik1 changed size and dynamic behaviour of the droplets (Fig. 1b and Supplementary Movies 2 and 3) but did not affect coalescence. Removing Kar9 fully solubilized the Bim1 + Bik1 mixture even at 200 mM salt: although Bik1 condensed on its own in reduced salt conditions (Supplementary Movie 4), Bim1 addition resolubilized it (Supplementary Movie 5). When alone, Kar9 was soluble only at 500 mM salt. At reduced salt, it precipitated without observed dynamics instead of condensing into liquid-like droplets (Supplementary Movie 6). Together, these data indicate that, while insoluble on its own, Kar9 promotes liquid–liquid phase separation of Bim1 and Bik1 at reduced salt, and is itself enriched and soluble within the droplets formed.

To better characterize the phase separation behaviour of the stoichiometric Kar9 + Bim1 + Bik1 mixture, we analysed it with varying salt and protein concentrations (150–425 mM NaCl and 0.1–10 µM of each protein). Droplet formation was screened by microscopy. The Kar9 + Bim1 + Bik1 droplets formed even at 425 mM salt (Fig. 1c,d). Without Bik1, the Kar9 + Bim1 droplets dissolved at salt concentrations above 300 mM salt. Removing Bim1 increased the Kar9 + Bik1 mixture's critical concentration for droplet formation at most salt concentrations. At 200 mM salt, the critical protein concentration was between 0.5 µM and 1.0 µM each for Kar9 + Bim1 + Bik1 and between 1.0 µM and 2.5 µM for both Kar9 + Bim1 and Kar9 + Bik1. Varying solely Kar9 concentration indicated that 1.0 µM was sufficient to induce droplet

formation in an equimolar mixture of 10 µM each of Bim1 and Bik1 (Extended Data Fig. 2b). Thus, even in absence of crowding, the core Kar9-network components underwent phase separation over a broad range of sodium chloride and protein concentrations.

Analysis of droplet size, morphology and dynamics within minutes from dropping the salt concentration indicated that those formed by Bik1 alone were the most fluid, fusing and relaxing within seconds, followed by Kar9 + Bim1 + Bik1 (tens of seconds) and Kar9 + Bik1 and Kar9 + Bim1 droplets (hundreds of seconds; Fig. 1b,e). Kar9 + Bim1 + Bik1 and especially Kar9 + Bim1 droplets became rapidly viscous; droplets joining late in the reaction formed chains faster than they could relax to round shape (Fig. 1b). Thus, droplet composition and ageing properties widely affected their fluidity.

### Kar9-network formation is governed by multivalency interaction

The data presented above indicated that Kar9, Bim1 and Bik1 must interact with themselves and each other through a multivalent network of binding interfaces. Several interactions between the core Kar9-network components Kar9, Bim1 and Bik1 have been previously determined. As illustrated in Fig. 2a, Kar9 is composed of an N-terminal spectrin-repeat-containing domain that binds Myo2 and mediates Kar9–Kar9 self-association[4]. The C-terminal domain of Kar9 is disordered and enriched with basic, serine and proline residues (Extended Data Fig. 3). It contains at least three linear motifs (SxIP and LxxPTPh), which bind the EB homology domain (EBH) of Bim1 (refs. [4,8,26,27]). Bim1 contains a microtubule-binding calponin homology (CH) domain that is connected to a dimerizing coiled coil and EBH domain by a basic- and serine-rich disordered linker[28] (Extended Data Fig. 3). The disordered C-terminal tail of Bim1 is acidic and terminates in an EEY/F motif that binds the cytoskeleton-associated protein-glycine-rich (CAP-Gly) domain of Bik1 (refs. [29,30]). In Bik1, the N-terminal CAP-Gly domain is connected to a dimerizing coiled coil domain by a basic- and serine-rich, disordered linker[31] (Extended Data Fig. 3). Its C-terminal portion contains a zinc-finger (ZnF) domain and a short, EEY/F-like motif. Interestingly, the homologues of Bim1 and Bik1 in mammals and fission yeast, that is, EB1 and CLIP-170, and Mal3 and Tip1, respectively, are found to also undergo phase separation in vitro and in vivo[32–34] (co-submitted with this manuscript). As Bik1 also undergoes phase separation on its own (Fig. 1b, ref. [35]), it must contain additional self-association interfaces besides the coiled-coil that are currently not known. Notably, the C-terminal domain of Kar9 also carries at least one binding site for Bik1 (ref. [23]), consistent with Kar9 phase separating together with Bik1 (Fig. 1b).

Since Kar9 is a strong driver of Kar9-network condensation (Fig. 1a,b and Extended Data Fig. 2b), we reasoned that it must contribute substantially to multivalency. Strikingly, the disordered C-terminal domain of Kar9 alone did not promote phase separation of Bim1 and Bik1 as much as full-length Kar9 (Fig. 2b). Therefore, and given its role in self-association, we investigated whether its N-terminal domain was important for multivalency and phase separation of the network.

The crystal structure of the Kar9 N-terminal domain from the budding yeast *Naumovozyma castellii* (NcKar9N) revealed three crystallographic dimers with total buried surface area per protomer of 1,549, 1,177 and 851 Å$^2$, respectively[4]. The interfaces forming these dimers are referred to as α, β and γ (Fig. 2c). While the α- and β-dimer interfaces make two symmetric contact points, the γ interface is established by a single one (Extended Data Fig. 4a–c). Remarkably, the residues forming the three interfaces (Methods) are well conserved across Kar9 orthologues (Extended Data Fig. 5).

To test whether any of the three self-association interfaces could mediate NcKar9N–NcKar9N interactions in vitro, we performed a mutagenesis study with His-tagged NcKar9N. To this end, we mutated two conserved residues in each of the interfaces to alanine (Fig. 2c and Extended Data Figs. 4a–c and 5). The integrity of the three mutants was

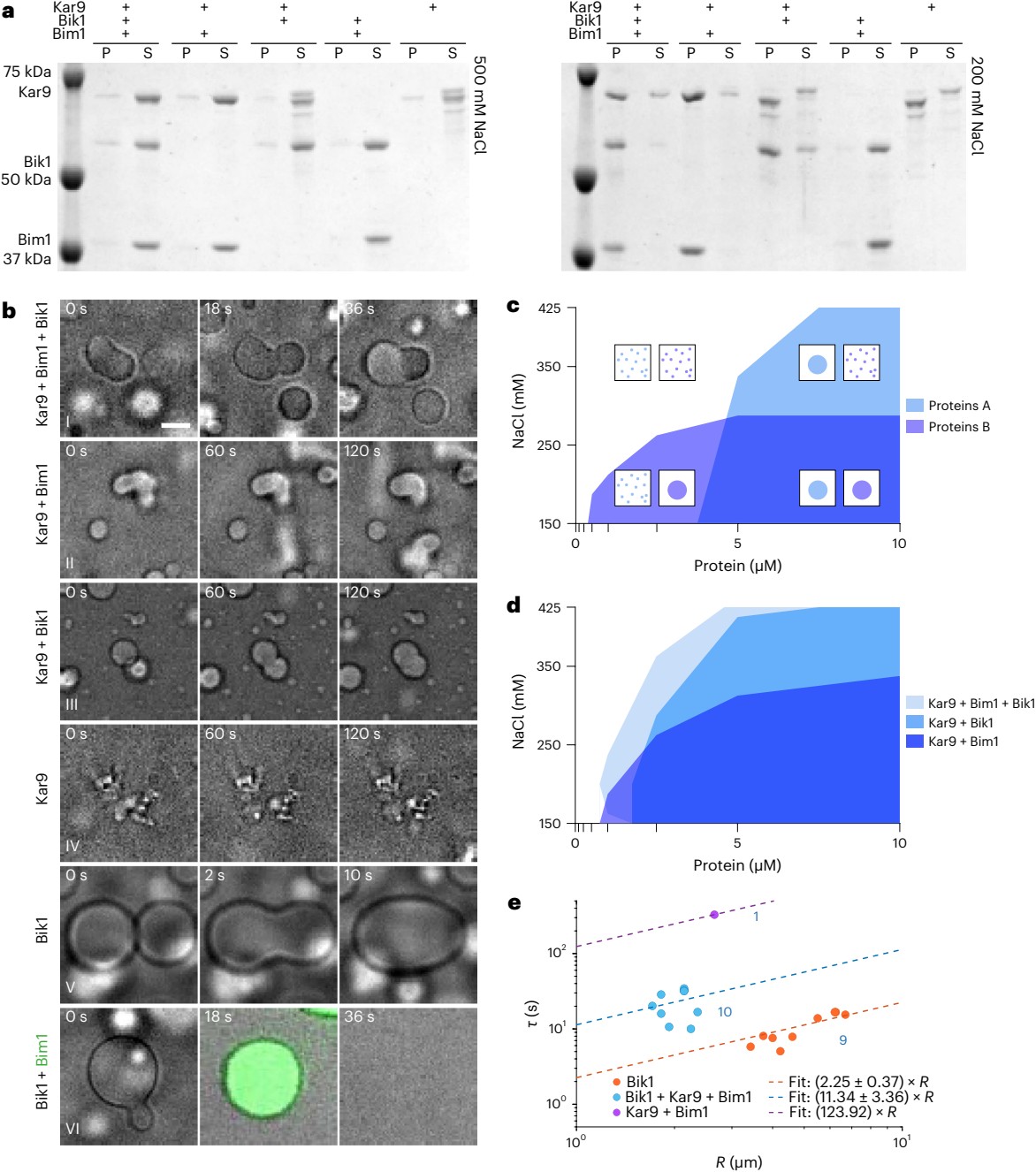

**Fig. 1 | Characterization of Kar9, Bim1 and Bik1 phase separation in vitro. a**, Pelleting assays of Kar9 and mixtures with Bim1 and Bik1 (10 μM each) analysed by Coomassie-stained SDS–PAGE at 500 mM (left) or 200 mM (right) NaCl. P, pellet; S, supernatant. For partitioning between light and dense phases, see Extended Data Fig. 2a. **b**, Time-lapse phase contrast micrographs of mixtures of proteins as indicated on the left. For Bik1 + Bim1, Bik1 droplets were pre-formed and then Atto-488-NTA-labelled Bim1 was added. Scale bar, 3 μm. See also Supplementary Movies 1–6. For additional FRAP data, see Extended Data Fig. 1. **c**, Schematic 2D superimposed phase diagram for separate experiments with imaginary proteins or protein mixtures A and B to compare their phase behaviour. Ticks on the axes indicate tested conditions, coloured areas the extrapolated regions with phase separation. Illustrations describe the local condensation states. **d**, Superimposed phase diagrams of equimolar mixtures of Kar9 + Bim1 + Bik1 (light blue), Kar9 + Bik1 (intermediate blue) and Kar9 + Bim1 (dark blue) obtained at different protein and NaCl concentrations. Protein concentration is the concentration of each protein in the equimolar mixture. Single experiment. For titration of Kar9 into constant Bim1 and Bik1, see Extended Data Fig. 2b. **e**, Droplet fusion time τ as a function of droplet radius R based on phase contrast microscopy movies (**b**) on a log–log plot. Dots indicate measurements; lines indicate linear fits to the data with reported slopes of τ as a function of R, blue values indicate n = number of droplets, single experiment. Details in Methods and Extended Data Fig. 10. Source numerical data and unprocessed gels available in source data.

confirmed by circular dichroism (CD) spectroscopy, which revealed spectra and cooperative thermal unfolding profiles similar to those obtained for the wild-type protein (Extended Data Fig. 4d,e).

The oligomerization state of the different NcKar9N variants was then assessed by size exclusion chromatography followed by multi-angle light scattering (SEC–MALS). Consistent with our previous results[4], analysis of wild-type NcKar9N (200 μM protein injected onto the size exclusion chromatography column) at 150 mM sodium chloride yielded molecular masses consistent with the presence of dimers (118 kDa; calculated molecular mass of wild-type His-NcKar9N

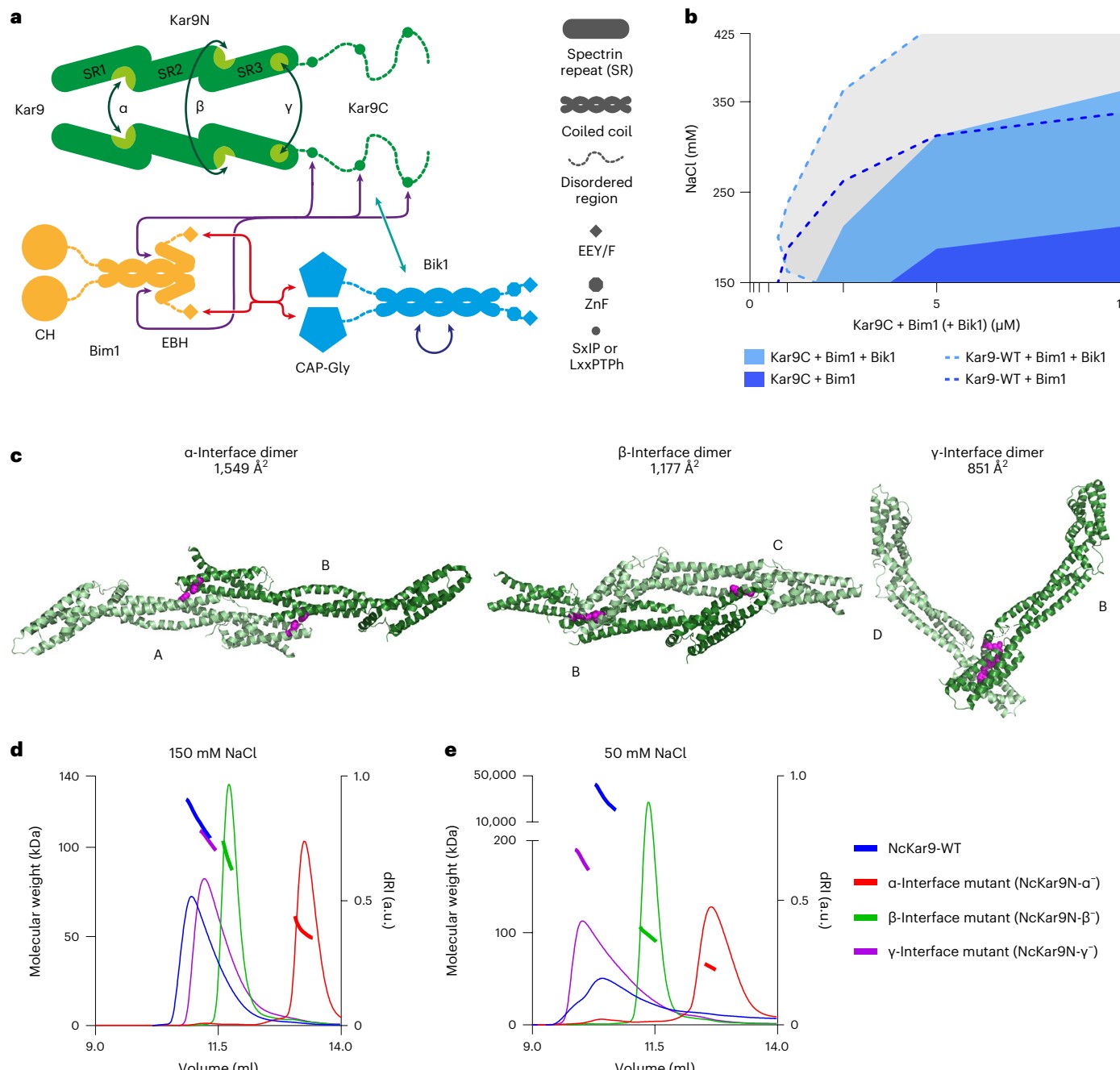

**Fig. 2 | Characterization of NcKar9N–NcKar9N interactions. a**, Schematic representation of the core Kar9-network module. Known protein–protein interactions between Kar9, Bim1 and Bik1 are indicated by double arrows. Additional information on disordered regions in Extended Data Fig. 3. **b**, Superimposed phase diagram of Kar9C with Bim1 in the presence (light blue) or absence (dark blue) of Bik1, compared with the phase diagrams of Kar9-WT + Bim1 + Bik1 (dashed light-blue line) and Kar9-WT + Bim1 (dashed dark-blue line) (Fig. 1d). Single experiment. **c**, The three crystallographic dimers interacting via interfaces α, β and γ in the NcKar9N crystal (PDB ID 7AG9) in ribbon representation. Colour distinguishes protomers; different protomers are denoted A, B, C and D. The residues mutated to disrupt each interface are represented as magenta spheres. **d**,**e**, SEC–MALS experiments of His-tagged NcKar9N (blue) and interface mutants NcKar9N-α⁻ (Phe288Ala/Phe344Ala; red), NcKar9N-β⁻ (Arg233Ala/Ile237Ala; green) and NcKar9N-γ⁻ (Tyr363Ala/ Arg364Ala; magenta) in the presence of 150 mM (**d**) or 50 mM (**e**) NaCl. See also Extended Data Fig. 4. For conservation of interfaces, see Extended Data Fig. 5. Source numerical data available in source data.

monomer is 50 kDa; Fig. 2d). At 50 mM salt, a broader elution profile was obtained, shifted towards a lower elution volume. The measured molecular mass increased to a value much higher than 200 kDa, suggesting the formation of non-stoichiometric oligomers (Fig. 2e). In both salt conditions, mutating the α-interface (NcKar9N-α⁻) resolved these multimers to monomers (52 kDa). The β-interface mutation (NcKar9N-β⁻) did not affect dimer formation at 150 mM salt but reduced the larger oligomers formed at 50 mM salt towards a molecular mass of a dimer (97 kDa). The γ-interface mutant (NcKar9N-γ⁻) reached a mass of 107 kDa at 150 mM salt and 180 kDa at 50 mM, consistent with limiting the formation of oligomers up to tetramers. These observations indicate that all three crystallographic contact sites of Kar9N mediate NcKar9N–NcKar9N interactions of different strengths in solution, with the α-interface mediating the most stable interaction.

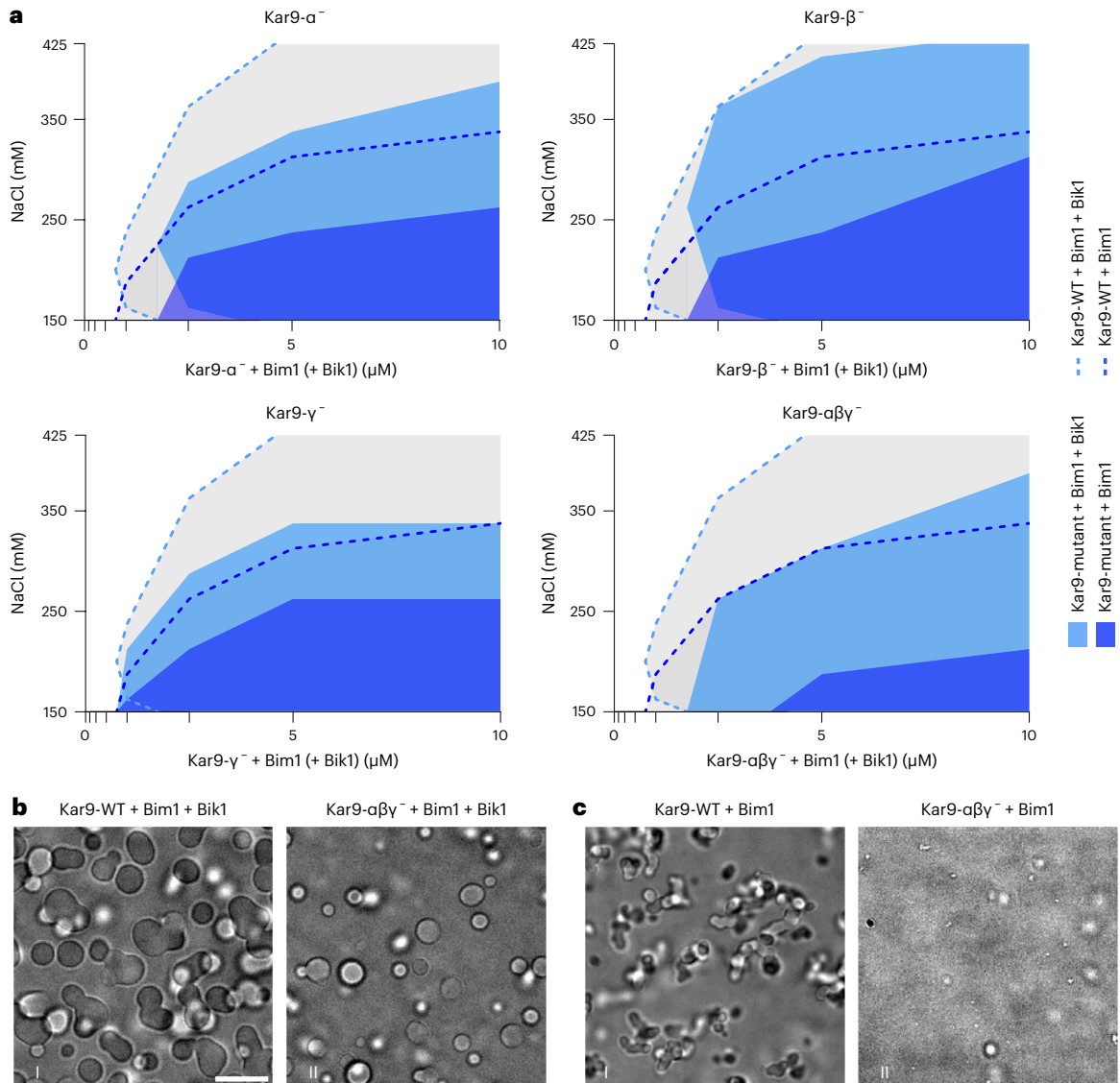

**Fig. 3 | Impact of mutations in core Kar9-network components on phase separation in vitro. a**, Superimposed phase diagrams of different Kar9-interface mutants with Bim1 in the presence (light blue) or absence (dark blue) of Bik1. For comparison, the phase diagrams of Kar9 + Bim1 + Bik1 (dashed light-blue line) and Kar9 + Bim1 (dashed dark-blue line) equimolar mixtures (Fig. 1d) are also shown in all plots. Single experiment. **b**,**c**, Phase contrast micrographs illustrating phase behaviour for the indicated protein mixtures and mutants at 220 mM NaCl and 10 μM protein. Scale bar, 10 μm. Source numerical data available in source data.

## Kar9 self-association drives phase separation of the Kar9 network

We next reasoned that the interactions of Kar9 with itself, as well as with Bim1 and Bik1 (Fig. 2a), probably drives phase separation of the Kar9 network in vitro. Furthermore, they should contribute cooperatively to this phenomenon. Therefore, mutating interaction sites individually may have some effect on the ability of the network to undergo phase separation. Accumulating such mutations should additively impair coalescence of network components. To test this idea, we inferred in *S. cerevisiae* Kar9 the self-association sites identified in NcKar9 on the basis of sequence conservation (Extended Data Fig. 5) and mutated them individually or in combination. We then investigated whether this affected the phase separation behaviour of the Kar9 network.

Interfering with any of the Kar9 self-association sites individually had some effect on the ability of the mutant protein to promote phase separation of the network (Fig. 3a): the corresponding phase diagrams deviated from that of wild-type Kar9 at 1.0 μM protein and at >300 mM

sodium chloride concentrations. Among the three Kar9 self-association interface mutants, Kar9-γ⁻ had the strongest effect at >300 mM salt and at high protein concentration (10 μM), followed by Kar9-α⁻. At <200 mM salt and lower protein concentrations (1 μM), Kar9-α⁻ showed the strongest effect. The Kar9-β⁻ mutant behaved very similarly to wild-type Kar9, except that it did not promote phase separation at lower protein concentration (1.0 μM). Mutating the three interfaces together enhanced these effects, although the mutant protein Kar9-αβγ⁻ still promoted phase separation of the protein mixture over a reduced but still broad range of concentrations and salinity (Fig. 3a,b). Strikingly, for each of these variants, Bik1 removal strongly reduced their ability to promote phase separation. The Bim1 + Kar9-αβγ⁻ mixture showed the strongest defect in coalescence (Fig. 3a,c).

Thus, Bik1 and the three Kar9 self-association interfaces contribute all cooperatively to the phase separation behaviour of the core Kar9-network components. As strong defects were observed only when combining Kar9 interface mutations and removing Bik1,

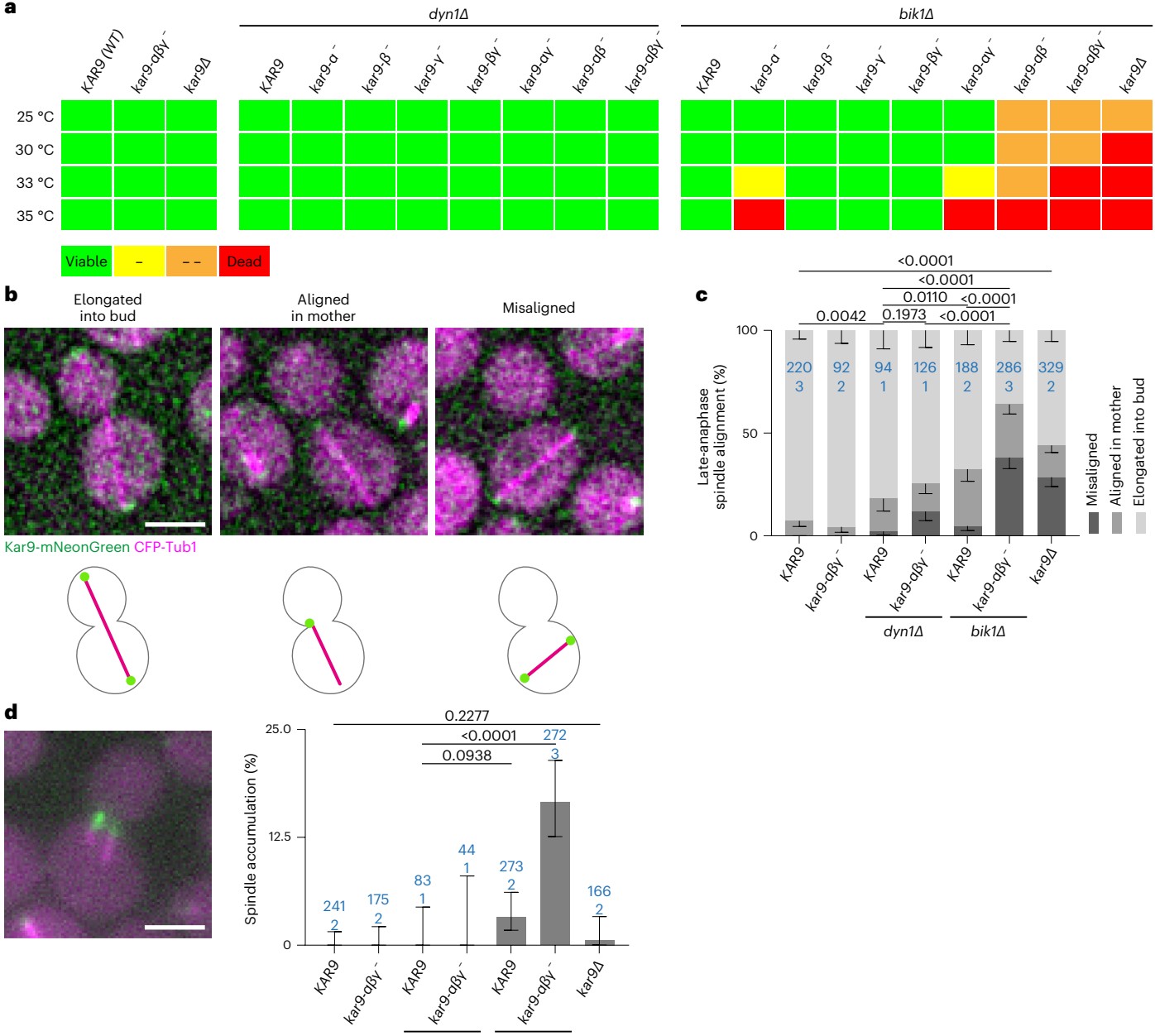

**Fig. 4 | Role of Kar9-network multivalency for cell division. a**, Summary of
viability spot assays of Kar9 self-association interface mutants combined with
deletion of *DYN1* or *BIK1* at different temperatures (see also Extended Data
Fig. 6). Viability is indicated by a colour code from green (viable) to red (inviable).
**b**, Deconvolved micrographs (top) and pictograms (bottom) showing late-
anaphase spindle (CFP-Tub1, magenta) positioning and alignment in cells
expressing wild-type Kar9 or interface mutants tagged with mNeonGreen (green)
at 25 °C. Scale bar, 3 µm. **c**, Frequency of late-anaphase spindle positioning and
alignment at 25 °C as shown in **b** (error bars are Wilson/Brown 95% confidence

intervals determined from binomial distribution). *P* values above bars on top
comparing frequency of spindles elongated to the bud were determined by two-
proportion *z*-test. **d**, Example micrograph and quantification of frequency of pre-
anaphase cells with more than one spindle at 25 °C (error bars are Wilson/Brown
95% confidence intervals determined from binomial distribution). *P* values are
determined by two-proportion *z*-test. Scale bar, 3 µm. Blue values in **c** and **d**
indicate *n* (cells analysed from number of biological replicates (clones) indicated
below). Source numerical data available in source data.

we concluded that the different homo- and heterotypic interactions
between Kar9 and Bik1 act redundantly in coalescence and that we
identified a sufficient number of interfaces to already substantially
perturb the process in vitro. Testing the role of the self-association
interfaces of Kar9 in absence of Bim1 proved not feasible because the
mutant forms of Kar9 became rapidly insoluble in absence of Bim1. Kar9
interacts with low nanomolar affinity with Bim1 (refs. [8,36]), suggesting
that it is always associated with Bim1 in living cells. Thus, for simplicity
the two proteins are considered as an obligate single unit for the rest of

this study, a notion that is consistent with the fact that Kar9 does not
localize to microtubules in absence of Bim1 (refs. [21,22]).

## Kar9-network multivalency facilitates spindle positioning in vivo

Next, we investigated whether the identified interfaces contribute to
Kar9-network function in vivo. We constructed mutant yeast strains
where the endogenous *KAR9* gene was replaced with alleles abrogat-
ing each Kar9 self-association interface alone, in combinations of two,

or all three together, in presence or absence of Bik1. As Kar9 function becomes essential for growth in cells lacking the dynein gene *DYN1* (ref. [37]), each *KAR9* allele was combined with the *dyn1Δ* mutation to assay their functionality. As beyond its role in the Kar9 network Bik1 is also essential for dynein function[37,38], we did not need to remove the *DYN1* gene in the *bik1Δ* mutant cells to assay Kar9 functionality.

As shown in Fig. 4a and Extended Data Fig. 6, the *kar9-αβγ⁻* triple interface mutant allele was lethal or nearly so in combination with *bik1Δ* at all tested temperatures (25–35 °C); in these cells the Kar9 network was not functional. Mutating any of the α, β or γ interfaces alone, or removing Bik1 individually, however, did not cause much of a phenotype at any of these temperatures, even when *DYN1* was deleted (Fig. 4a and Extended Data Fig. 6). Likewise, any combination of two of these mutations failed to cause a much detectable phenotype, except when the α interface and Bik1 were both absent, which affected cell growth at high temperature. We conclude that all three Kar9 self-interaction interfaces are functional in vivo, and that they work cooperatively with each other and Bik1.

To characterize how the progressive reduction of multivalency in the Kar9 network affected cell viability, we investigated the effects of cumulating mutations on spindle alignment and positioning (Fig. 4b,c). While *bik1Δ* and Kar9 triple-interface mutations separately had no or little effect on the outcome of mitosis at 25 °C, their combination caused the accumulation of late-anaphase cells with elongated spindles that remained in the mother cell and failed to segregate an SPB to the bud. Furthermore, two-thirds of the late-anaphase *kar9-αβγ⁻ bik1Δ* double-mutant cells mispositioned their spindles. Accordingly, up to 17% of these mutant cells contained more than one spindle during mitosis, a phenotype that is essentially never observed in wild-type cells (Fig. 4d).

### Multivalency underlies Kar9 restriction to few microtubules in vivo

Thus, we next investigated whether multivalency contributes to Kar9 protein localization in vivo. At 25 °C, cells expressing wild-type or Kar9 mutant proteins tagged with the fluorescent protein mNeonGreen revealed that mutating the γ interface had little effect, but erasing the α or β interface individually reduced pre-anaphase Kar9 levels at microtubule plus-ends by 40% and 20%, respectively (Fig. 5a). Simultaneously mutating an increasing number of Kar9 self-association interfaces further decreased Kar9 levels at microtubule ends. Abrogating all three interfaces reduced these levels by 60% and by 70% when Bik1 was removed as well. Strikingly, the Kar9-αβγ⁻ protein failed to remain focused on the tip of a single microtubule. Rather, this mutant protein distributed more symmetrically to microtubules on both sides of the spindle compared with its wild-type counterpart (asymmetry index of 0.75 and 0.9, respectively; Fig. 5b). This phenotype was enhanced upon *BIK1* deletion (0.42 in the *kar9-αβγ⁻ bik1Δ* double-mutant cells; Fig. 5b). In these cells, the reduced Kar9 levels at microtubule plus-ends were not only due to the redistribution of the protein to several

microtubules since the total protein amount localizing to microtubules was still very low compared with wild-type and single-mutant cells (Extended Data Fig. 7).

### Multivalency enables Kar9 to track shrinking microtubule ends

The Kar9 network possesses the unique ability to track both growing and shrinking microtubule plus-ends[19,20]. We reasoned that multivalency might support this process by increasing the avidity of the Kar9 network for microtubule tips. Thus, we imaged the wild-type and Kar9-αγ⁻ mutant proteins tagged with mNeonGreen at high spatial and temporal resolution. As previously shown[20], the punctum formed by wild-type Kar9 tracked the plus-end of its target microtubule throughout its growth and shrinkage cycles (Fig. 5c,d and Supplementary Movie 7). Other Kar9 puncta formed occasionally on tips of other microtubules but rapidly disappeared either by fusing with the main punctum or by fading away in the cytoplasm.

This persistent tracking of growing and shrinking microtubule plus-ends by Kar9 was occasionally lost in the *kar9-αβγ⁻* and *bik1Δ* single-mutant cells (Fig. 5d,e and Supplementary Movie 8). This phenotype was synergistically enhanced in the *kar9-αβγ⁻ bik1Δ* double-mutant cells: nearly 50% of them lost their main Kar9 punctum within 3 min of imaging (9% in wild type; Fig. 5d,e). Puncta tended to then re-assemble next to SPBs and track the next emerging microtubule plus-end. Punctum disappearance generally happened at what seemed to be the end of a microtubule growth phase or shortly after microtubule shrinkage started.

To quantitatively analyse whether Kar9 localization depends on microtubule length changes, we automatically identified microtubule growth and shrinkage phases (Extended Data Fig. 8a; for details, see Methods). Rates estimated from individual phases suggested distinctions between wild type and mutants (Extended Data Fig. 8b,c), but they were not directly comparable because durations of shrinkage phases differed significantly between strains (Extended Data Fig. 8d). We therefore estimated linear mixed effects models jointly for the strains, because they capture the behaviour of an 'average' microtubule, as well as variability between microtubules (Methods). This analysis indicated that microtubule growth and shrinkage rates did not differ between strains (Fig. 5g; note significantly different initial lengths due to different phase durations and thereby overall dynamics, Extended Data Fig. 8e). In fitting with the loss of Kar9 puncta reported above, in the *kar9-αβγ⁻ bik1Δ* double-mutant cells the average intensity of the Kar9 punctum decreased during microtubule shrinkage, unlike in wild-type cells where it rather increased (Fig. 5f and Extended Data Fig. 8c). Again, this behaviour was not observed nearly as clearly in cells carrying either *bik1Δ* or Kar9 triple interface mutations alone.

### Kar9 structures at microtubule ends are non-stoichiometric

As Kar9 function in vivo (Figs. 4 and 5) correlates extremely well with Kar9 ability to promote condensation of the network components in vitro (Figs. 1–3), we hypothesized that the network proteins might

**Fig. 5 | Dissection of Kar9-pathway defects caused by reduced Kar9-network multivalency. a**, Kar9-mNeonGreen fluorescence intensity levels in Kar9 self-association interface mutants and *bik1Δ* at 25 °C at a single astral microtubule plus-end (normalized median with 95% confidence intervals). *P* values determined by Welch's *t*-test. For cumulative fluorescence intensity on all plus-ends, see Extended Data Fig. 7. **b**, Kar9 asymmetry index (Methods) quantifying Kar9-mNeonGreen distribution between astral microtubule tips on both sides of metaphase spindles at 25 °C (median with 95% confidence interval). *P* values determined by Welch's *t*-test. Example for a weakly symmetric *kar9-αβγ⁻ bik1Δ* cell with labels for quantified intensities is shown on the right. Scale bar, 3 μm. Blue values in **a** and **b** indicate *n* = cells analysed/number of biological replicates. **c**, Micrographs showing Kar9-mNeonGreen tracking the plus-end of a shrinking astral microtubule in wild type (WT, arrowheads mark the period of shrinkage) and failure to do so in a *kar9-αβγ⁻ bik1Δ* cell at 25 °C (the asterisk indicating loss of Kar9 punctum). Scale bars, 3 μm and 1 μm (in zooms). See also Supplementary

Movies 7 and 8. **d**, Astral microtubule length profiles for wild-type, *kar9-αβγ⁻*, *bik1Δ* and *kar9-αβγ⁻ bik1Δ* cells (moving average over three timeframes of distance between Kar9-mNeonGreen and Spc42-mCherry). Profiles persisting for the whole length of the movie are displayed in grey; profiles that end owing to loss of punctum (indicated with arrowhead) in red. **e**, Frequency of cells losing Kar9-mNeonGreen punctum on astral microtubule plus-ends over 175 s (error bars are Wilson/Brown 95% confidence intervals determined from binomial distribution). *P* values are determined by two-proportion *z*-test. Blue values indicate *n* (cells analysed, single experiment). **f,g**, Estimated rates of Kar9-mNeonGreen fluorescence change (**f**) and astral microtubule length change (**g**) for growing (blue) and shrinking (red) microtubules. Symbols indicate estimates and lines their 95% confidence intervals. Colour of *P* values (two-sided *t*-test) and *n* (number of timeframes considered for the model, single experiment) matches the growth/shrinkage category they refer to. For details, see Methods and Extended Data Fig. 8. Source numerical data available in source data.

function together by forming a condensate at the tip of target microtubules in vivo. However, in vivo Kar9 puncta are smaller than the diffraction-limited resolution of light microscopy, preventing the visualization of their morphology and direct assay of their material properties. Instead, we investigated whether they showed other hallmark properties of biomolecular condensates. One such property

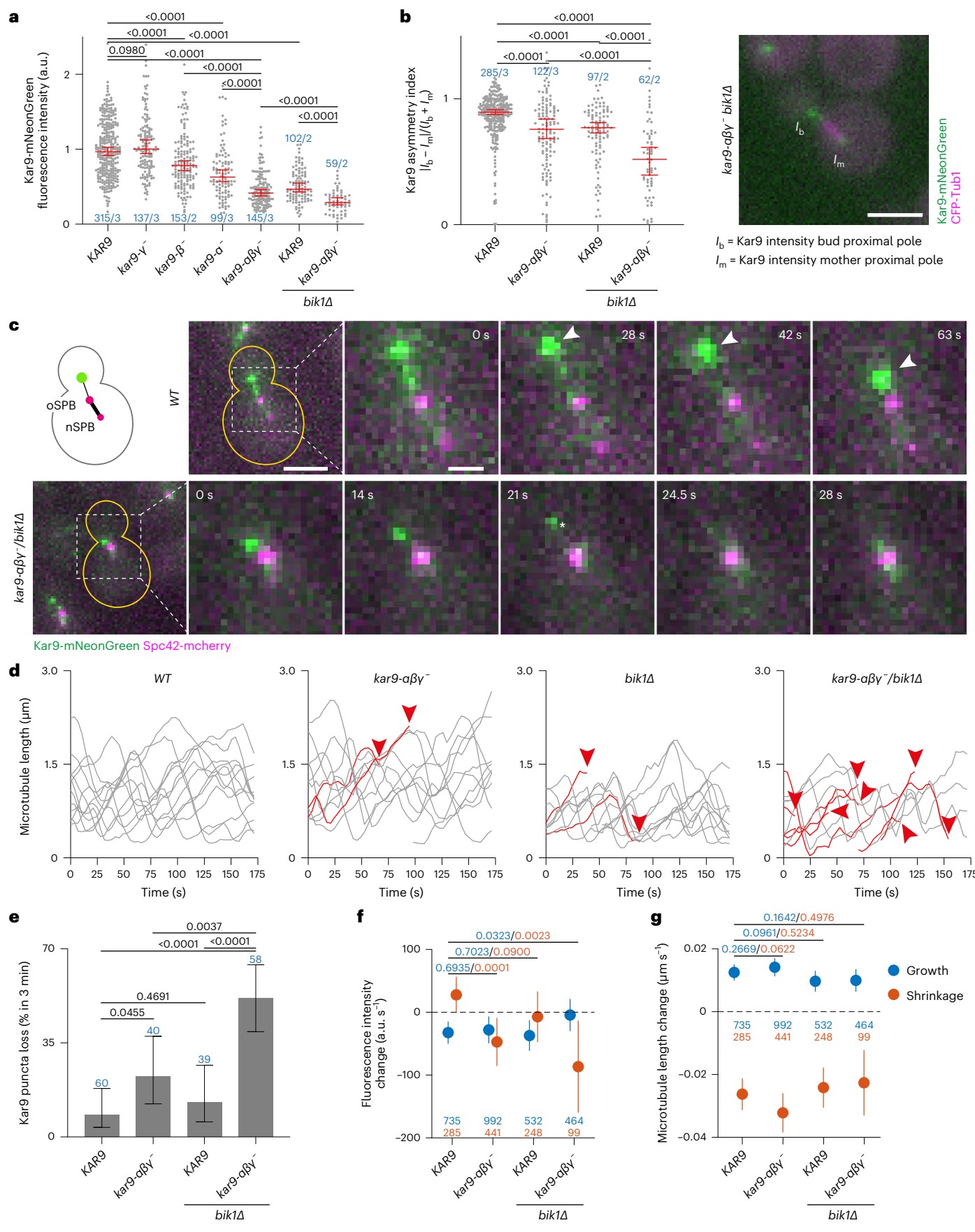

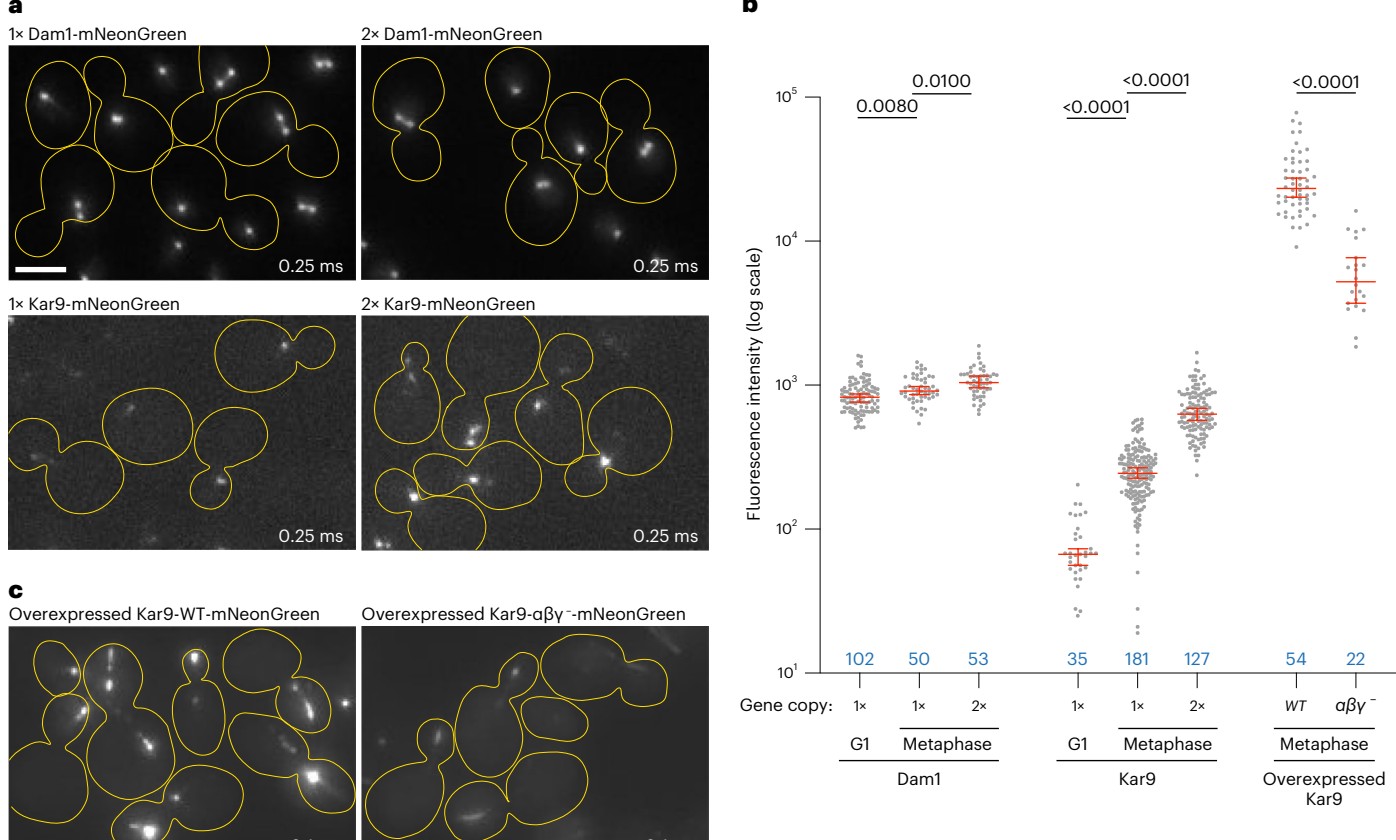

**Fig. 6 | Kar9 stoichiometry at microtubule plus-ends. a**, Micrographs showing Dam1-mNeonGreen or Kar9-mNeonGreen localization in haploid cells bearing one (1×) or two (2×) fluorescently tagged gene copies at 25 °C (exposure time indicated on the bottom right). Scale bar, 3 µm. **b**, Fluorescence intensity levels of Dam1-mNeonGreen and Kar9-mNeonGreen during G1- and metaphase, of the cells described in **a** (median with 95% confidence interval; *P* values determined by Welch's *t*-test). Blue values indicate *n* (cells analysed, single experiment). **c**, Micrographs showing the localization of Kar9-mNeonGreen overexpressed from the galactose promoter in haploid cells (note the difference in exposure time). Source numerical data available in source data.

consists of their ability to vary in size as the amount of available material changes (reviewed in ref. [39]). Thus, we asked whether Kar9 puncta showed such behaviour in vivo.

Quantification of the fluorescence of the Kar9-mNeonGreen puncta showed that they widely varied in intensity as cells progressed from G1 to metaphase, upon insertion of a second copy of the gene in the genome, or when overexpressed (Fig. 6a–c). Abrogating all three Kar9 self-association interfaces extensively reduced puncta intensity upon overexpression (Fig. 6b,c), as for the endogenous protein (Fig. 5a). In contrast, the Dam1 protein, which forms a stoichiometric complex at the tip of kinetochore microtubules[40,41], formed puncta of constant fluorescence intensity when tagged with mNeonGreen, irrespective of cell cycle stage or Dam1 expression levels (Fig. 6a,b). Thus, unlike Dam1, the Kar9 body is not a stoichiometric structure, but a condensed, cohesive structure. Dam1 is estimated to be present in 17 copies per kinetochore microtubule[40]. Using its intensity as a reference, we estimate that metaphase cells recruit, on average, at least 70 Kar9 molecules at the tip of a single microtubule.

**Kar9 shows a liquid-like behaviour on astral microtubule ends**

As mentioned earlier, in cells containing several Kar9 puncta, these frequently merged with each other (Fig. 7a,b and Supplementary Movie 9) and stably remained together for many subsequent time-frames, suggesting that they had truly fused into a single entity. In wild-type cells, such fusion events took place every 2.6 min on average, whereas fission events were two-fold less frequent (Fig. 7b), as

expected given the fact that droplet fusion is energetically more favourable than fission. Decreasing the multivalency of the system increased both rates and reduced their relative ratio. These observations suggest that the Kar9 network functions as a liquid-like condensate in vivo.

Hexanediol resolubilizes certain liquid–liquid phase separated proteins both in vitro and in vivo by disturbing low-affinity, hydrophobic interactions, but leaves aggregates and stoichiometric complexes relying on high-affinity interactions mostly intact[42]. In vitro, phase separation of equimolar mixtures of Kar9 + Bim1 + Bik1 droplets was indeed severely affected by addition of 5% hexanediol, especially at intermediate sodium chloride concentrations (>250 mM; Fig. 7c). Thus, we tested whether Kar9 puncta intensity was also sensitive to hexanediol in vivo. Imaging pre-anaphase cells immediately after resuspending them in medium containing 5% hexanediol reduced puncta intensity about eight-fold (Fig. 7d,e). Both mNeonGreen-Bim1 and mNeonGreen-Bik1 intensities at astral microtubule tips were similarly affected (Fig. 7e and Extended Data Fig. 9a). Hexanediol did not act indirectly by depolymerizing astral microtubules since the treatment had only a mild effect on them, as determined using GFP-Tub1 as a reporter (Extended Data Fig. 9b,c). Furthermore, eliminating all astral microtubules using nocodazole did not severely affect the intensity of Kar9-mNeonGreen puncta that re-localized close to SPBs in these cells (Extended Data Fig. 9d,e). Thus, we concluded that hexanediol potently and rapidly inhibited the liquid–liquid phase separation of the Kar9-network components to similar extents in vitro and in vivo.

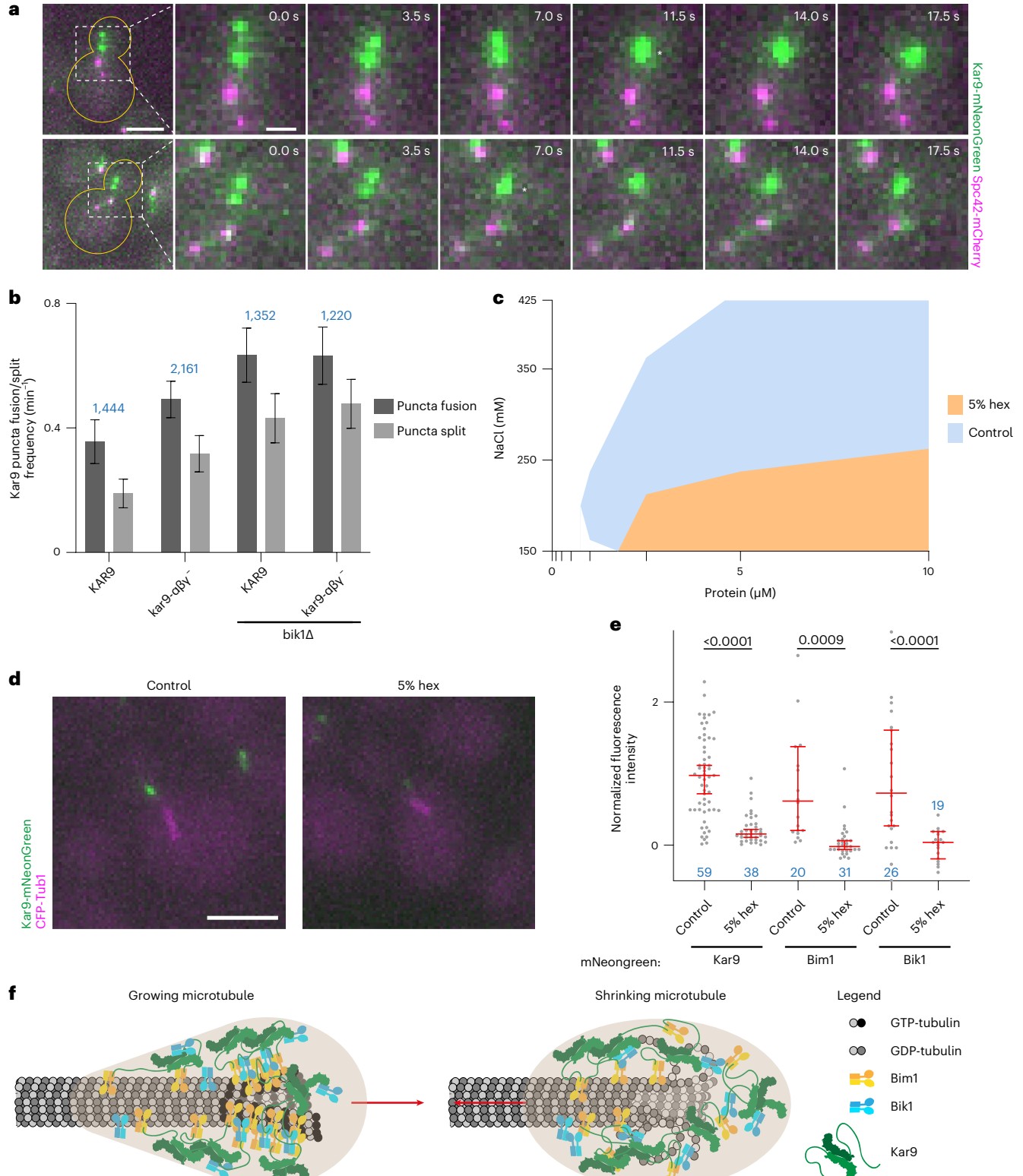

**Fig. 7 | Kar9 +TIP-body liquid-like behaviour in vivo. a**, Micrographs showing fusion events between Kar9-mNeonGreen puncta. Scale bars, 3 μm and 1 μm (in zooms). See also Supplementary Movie 9. **b**, Kar9 puncta fusion/split frequencies in cells analysed in Fig. 5c–g. Blue values indicate *n* (number of analysed timeframes, single experiment). **c**, Superimposed phase diagrams comparing the phase separation of 1:1:1 mixture of Kar9 + Bim1 + Bik1 in presence or absence of 5% 1,6-hexanediol (hex). Single experiment. **d**, Micrographs of a Kar9-mNeonGreen expressing control cell and cell treated with 5% 1,6-hexanediol immediately before acquisition at 25 °C. Scale bar, 3 μm. For micrographs of Bim1 and Bik1, see Extended Data Fig. 9a. **e**, Kar9-mNeonGreen, mNeonGreen-Bim1 and mNeonGreen-Bik1 fluorescence intensity levels in wild-type cells treated with 5% 1,6-hexanediol versus untreated control (normalized median with 95% confidence intervals). Significance levels determined by Welch's *t*-test. Additional data about effect on astral microtubules in Supplementary Fig. 9b–e. Blue values indicate *n* (cells analysed, single experiment). **f**, Proposed model for how the +TIP body tracks growing (left) and shrinking (right) microtubules. Source numerical data available in source data.

## Discussion

A wide range of biomolecules use low-affinity, multivalent interactions to undergo liquid–liquid phase separation and generate membrane-less organelles (reviewed in refs. [39,43]). Here we show that a dense web of redundant interactions links the core components of the Kar9 network together to assemble a nanometre-scale structure, which we name '+TIP body', at the plus-end of essentially one single cytoplasmic microtubule in yeast. Our data suggest that this body is a liquid-like condensate, its dynamic and microtubule-binding properties allowing it to persist and track both the growing and shrinking plus-end of its target microtubule. We postulate that the material properties of the nanometre scale Kar9 +TIP body supports its function as a 'mechanical coupling device' that supports the transfer of forces between the actin and the microtubule cytoskeleton to properly position the mitotic spindle during cell division.

Several lines of evidence support our hypothesis. Firstly, phase separation in vitro relies on multiple high-, medium- and low-affinity interactions within the Kar9 network. The identified interfaces (Kar9–Kar9 (ref. [4] and this study) and Kar9–Bim1 (refs. [8,36])) are all conserved across Kar9 homologues and are therefore all functionally relevant. Accordingly, these interaction interfaces are instrumental for proper function of the Kar9 network. Furthermore, their redundancy in vivo perfectly parallels their cooperativity for network phase separation in vitro. Secondly, the Kar9 +TIP body forms a non-stoichiometric assembly, able to undergo fission and fusion, supporting the idea that it is a liquid-like structure within the cell. Finally, the proper function of this assembly does not depend on the exact amount of Kar9 protein present at the microtubule tip. Indeed, the substantial reduction of Kar9 levels caused by most single and double mutations of the self-association interfaces does not translate into noticeable functional defects. Thus, our data indicate that the Kar9 network undergoes condensation in cells and that this property is instrumental to fulfil its cellular function.

Considering that the phenotypic consequences of decreasing multivalency in the system indicates which of the body's functionalities are specifically dependent on phase separation, our data provide compelling support for condensation mediating at least two key functional properties of the network. Firstly, condensation underlies recruitment and concentration of Kar9 to a selected microtubule plus-end in cells. Kar9 has a high affinity for Bim1, and Bim1 for the plus-ends of growing microtubules[8,26]. Thus, its recruitment to microtubule plus-ends via Bim1 must locally enhance Kar9 concentration and may nucleate condensation, leading to recruitment of more Kar9 and its partners. Ostwald ripening of this condensate, that is, the tendency of the biggest condensate to collect all the available material, could explain why budding yeast cells form virtually only one Kar9 +TIP body[20]. Thus, condensation offers a determining mechanism to specialize the underlying microtubule.

Secondly, multivalency underlies the remarkable capacity of the Kar9 +TIP body to track shrinking microtubule ends. As condensation proceeds, accruing Kar9 levels should bring an excess of Bim1 molecules with itself, allowing Bim1 binding to low-affinity sites on the microtubule lattice[44]. Supporting this notion, we found that overexpression of Kar9 tends to extend the +TIP body along the microtubule lattice (Fig. 6c). We suggest that such lattice 'wetting' supports both persistence of the +TIP body on shrinking microtubule plus-ends (Fig. 7f), as well as its cohesion to the dynamic microtubule plus-end necessary to support the transmission of pulling forces generated either by microtubule shrinkage or by directional movement of myosin motors along actin cables. Thereby, condensation facilitates the function of the +TIP body in mechanically linking microtubules and actin cables to mediate nuclear positioning.

The hypothesis of a liquid-like Kar9 +TIP body is especially attractive because it helps explaining cellular observations that remained difficult to rationalize so far, such as Kar9's unique localization pattern and its remarkable role in 'gluing' together two highly dynamic intracellular structures, namely microtubule tips and actin cables. Similar mechanisms may also explain how other patterning +TIPs, such as APC and MACF, target, persist and specialize the dynamics and functions of specific subclasses of microtubules in other cell types. Importantly, three recent studies by Miesch et al.[32], Song et al.[34] and Maan et al.[33] strongly support our overall conclusion that +TIPs can form nanometre-scale phase-separated bodies around microtubule ends. Investigating further the condensate properties of +TIP bodies may thus change our current view and understanding of how +TIPs accumulate to selected microtubule plus-ends, functionally specialize them, control their dynamics and regulate their interactions with different cellular targets.

## Online content

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

## Methods

### Cloning and protein preparation

**Plasmid preparation.** Expression vectors were produced either by homologous recombination with polymerase chain reaction (PCR)-amplified or synthetic genes and amplified backbones or site-directed mutagenesis (QuikChange, Agilent). The wild type[4] and mutant N-terminally hexa-histidine-tagged *N. castelli* Kar9 N-terminal domain (NcKar9N, residues 1–410) were cloned into the pET-based expression vector PSPCm2 (ref. [45]). The interface mutations in *N. castellii* Kar9 (UniProt ID G0VE12) were created by QuikChange and are as follows: α, Phe288Ala/Phe344Ala; β, Arg233Ala/Ile237Ala; γ, Tyr363Ala/Arg364Ala. Full-length wild-type[8] and mutant and Kar9C (405–644) N-terminally hexa-histidine-tagged *S. cerevisiae* Kar9 (UniProt ID P32526) were cloned into the Acembl vector pACE[46]. The interface mutations in *S. cerevisiae* Kar9 were created in yeast (see below) and then cloned into pACE. They are as follows: α, Phe292Ala/Leu347Ala; β, Arg237Ala/Asn241Ala; γ, Tyr366Ala/Arg367Ala. Full-length *S. cerevisiae* wild-type and N-terminally hexa-histidine-tagged *S. cerevisiae* Bim1 (ref. [8]) (UniProt ID P40013), and *S. cerevisiae* Bik1 (UniProt ID P11709) were cloned into pET3d (Invitrogen) and PSTCm1 (ref. [45]), respectively. All primers are listed in Supplementary Table 1.

**Protein expression and purification.** Sequence verified plasmids were transformed into BL21(DE3) *E. coli* cells for protein expression. To produce the proteins, liquid cultures were shaken in LB medium containing the appropriate antibiotic until an optical density at 600 nm (OD$_{600}$) of 0.6 was reached. Expression was induced by the addition of 0.75 mM isopropyl-β-D-thiogalactoside. Induced cells were further incubated overnight at 20 °C. Cells were lysed using the sonication method or high-pressure homogenizers (Avestin Emulsiflex-C3 High Pressure Homogenizer or Microfluidics Microfluidizer) in 20 mM Tris–HCl (pH 7.5), supplemented with 800 mM (full-length Kar9 alone[8]) or 500 mM (all other proteins) NaCl. Cell debris was removed by centrifugation. Cleared cell lysates were filtered using a 0.45 μm filter. Affinity purification of His-tagged proteins was carried out at 4 °C by immobilized metal affinity chromatography on Ni$^{2+}$-Sepharose columns (Cytiva) according to the manufacturer's instructions. For NcKar9N wild-type and full-length Kar9 (purified without Bim1), the hexa-histidine tag was enzymatically cleaved with PreScission and TEV protease, respectively. All protein samples were further applied on a pre-equilibrated Superdex-75 or −200 size exclusion chromatography column (Cytiva). For the production of different Kar9-Bim1 complex variants, wild-type or mutant hexa-histidine-tagged Kar9 and untagged Bim1 were separately expressed in bacteria and mixed before lysis in 500 mM NaCl. The purity of recombinant proteins was confirmed by Coomassie-stained SDS–PAGE, and the identities of the proteins were assessed by mass spectral analyses.

Despite substantial efforts, we were unsuccessful in purifying full-length Kar9 oligomerization interface mutants in absence of Bim1. Purification of full-length Kar9 without Bim1 was also challenging in case of wild-type protein. Most likely, in absence of Bim1 the hydrophobic SxIP and LxxPTPh Bim1 binding motifs unspecifically stick to other parts of Kar9 or cellular debris during purification, leading to low yield. We can only speculate, why Kar9–Kar9 oligomerization interface mutations made the situation even worse. Potentially, even more sticky patches were exposed in absence of oligomerization, reducing overall solubility even further.

### In vitro phase separation assays

**Pelleting assay.** Protein stocks were mixed and diluted in 1.5 ml test tubes with 20 mM Tris–HCl, pH 7.5, supplemented with 500 mM NaCl to reach a protein concentration of 40 μM each in 5 μl total volume at 500 mM NaCl (722 mM for Kar9 and Kar9 + Bik1). Next, 15 μl of low (100 mM or 26 mM for Kar9 and Kar9 + Bik1) or high-salt (500 mM or 426 mM for Kar9 and Kar9 + Bik1) Tris buffer was added to dilute the samples 4:1 to reach a final concentration of 10 μM in 200 or 500 mM

NaCl, respectively. The mixtures were then incubated in a thermomixer for 30 min at 25 °C while gently shaking. After 10 min spinning at 16,900*g* and 25 °C, the supernatant was transferred into a separate tube. The invisible pellets were resuspended in 20 μl of the Tris buffer (containing either 200 or 500 mM NaCl). Samples were analysed by Coomassie-stained 10% SDS–PAGE.

**Static light-microscopy assay for the assembly of phase diagrams.** A dilution series of either 5× (wild-type proteins) or 4× (mutant proteins and Kar9 + Bik1, due to concentration limitations) protein mixtures as above was prepared at high salt (usually 500 mM, except experiments with Kar9 purified alone 718 mM) to reach a broad range of protein concentrations (usually 0.1–10 μM final concentrations after dilution, thus either 0.5–50 or 0.4–40 μM before dilution). The reservoir wells of an MRC2 crystallization plate (Swissci) were then filled with buffers matching the final salt concentration of 150–425 mM NaCl. Next, the protein stock solutions were dispensed into the MRC2 plate's drop wells and diluted 5:1 or 4:1 (depending on the protein stock concentration) in a total volume of 1 μl using a Mosquito pipetting robot (SPT Labtech) with dilution buffers provided in a different plate to reach salt concentrations matching the reservoir wells. After 30 min incubation, all wells were imaged by phase contrast microscopy. The phase separation state of each well is reported in the source data for the phase diagrams shown in Figs. 1, 2, 3 and 7 and Extended Data Fig. 2. An approximate binodal line was constructed between phase-separated and soluble conditions (through the middle points at each tested protein concentration).

**Time-resolved microscopy assay.** Either 5× or 4× protein mixtures were prepared and then diluted to 10 μM and 220 mM NaCl in a Nunc Lab-Tek II chambered eight-well coverglass. To verify enrichment of His-tagged protein in droplets, Atto-488-NTA was used at 1 μg ml$^{-1}$ concentration. Samples were imaged on a DeltaVision Personal microscope (Cytiva) at 60× or 100× magnification using transmission and FITC channels in a single plane close to the coverglass surface.

**Droplet fusion analysis.** In the inset of the Extended Data Fig. 10, a typical fusion event of Bik1 is shown. To measure the fusion time $\tau$, we assume that a uni-axially deformed droplet relaxes to its final spherical state.

Parameter $= \frac{L-W}{L+W}$, was used to calculate the fusion time $\tau$.

$L$ is the major axis of the deformed droplet, and $W$ is the minor axis. Fitting an exponential curve to the time-dependent parameter $A$ would give the fusion time $\tau$ (Extended Data Fig. 10).

$\tau$ has relationship with viscosity and surface tension via Navier–Stokes equation:

$$\tau = \frac{(2\lambda + 3)(19\lambda + 169)}{40(\lambda + 1)} \frac{\eta_{\text{ext}} R}{\gamma} \tag{1}$$

where $\lambda = \eta_{\text{int}}/\eta_{\text{ext}}$ is the ratio of internal and the external viscosities, $R$ is the final radius of the droplet in spherical state and $\gamma$ is the surface tension.

In the case that $\lambda \ll 1 (\eta_{\text{int}} \ll \eta_{\text{ext}})$, one can simplify the equation to:

$$\tau = \frac{19}{20} \frac{\eta_{\text{int}} R}{\gamma} \tag{2}$$

which is valid where the viscosity of the protein condensed phase ($\eta_{\text{int}}$) is much higher than the viscosity of the protein dilute phase ($\eta_{\text{ext}}$). For instance, viscosity of the Bik1 droplet phase has been reported previously as $\eta_{\text{int}} \approx 18.2 \, (Pa \cdot s)$ which is four orders of magnitude higher than the Bik1 dilute phase[35].

In Fig. 1e, $\frac{\tau}{R} = \frac{19}{20} \frac{\eta_{\text{int}}}{R}$ (equation (2)) the slope is measured from fitting the data linearly. The value of $\frac{\tau}{R}$ for Bik1 is measured as $2.25 \pm 0.37 \, (s/m)$, which is consistent with the previously reported values of viscosity and surface tension for Bik1, 18.2 $(Pa \cdot s)$ and 7 (μN m$^{-1}$), respectively[35].

However, higher values of $\frac{\tau}{R}$ for Bik1 + Kar9 + Bim1 and Kar9 + Bim1, $11.34 \pm 3.46$ and $123.92\,(s/m)$, suggests much slower fusion for these systems. Unfortunately, owing to the slow fusion speed and the comparatively fast chain formation, we could resolve only one isolated two-droplet fusion event. Therefore, there is no meaningful error reported for this event[47,48].

**Droplet partitioning and FRAP.** Samples were prepared as for time-resolved microscopy above but at only 5 µM protein concentration (to reduce number of formed droplets) and without Atto-488-NTA. Ten-fold less concentrated mNeonGreen-Bik1, Kar9-GFP or Bim1-GFP were added to the mixture of unlabelled proteins to study the dynamics of all three components separately. Fluorescence images (partitioning) and movies (FRAP) were acquired with a Visitron Spinning Disk microscope in a single confocal slice in GFP channel close to the bottom of the well with 1 s time intervals. For FRAP, droplets were fully bleached by a laser pulse after timeframe 5 and recovery was observed for 3 min. For partition coefficient calculation, background (unlabelled protein mixture)-subtracted integrated densities were determined inside and outside droplets using Fiji and the inside/outside ratio was calculated. For FRAP quantification, background-subtracted integrated densities were determined using Fiji. For Bik1 droplets, bleaching/defocus was corrected by dividing by the average normalized intensity of nearby reference droplets.

**Biophysical characterization**
SEC–MALS experiments were performed in 20 mM Tris–HCl, pH 7.5, supplemented with 150 or 50 mM NaCl, 1 mM dithiothreitol using a S-200 10/300 analytical size exclusion chromatography column (GE Healthcare) connected in-line to mini-DAWN TREOS light scattering and Optilab T-rEX refractive index detectors (Wyatt Technology). Thirty microlitres of His-NcKar9N protein variants at a concentration of 200 µM were injected for each run onto the size exclusion chromatography column.

CD spectra of His-NcKar9N protein samples were recorded at 5 °C and at a protein concentration of 0.2 mg ml$^{-1}$ in 20 mM Tris–HCl, pH 7.5, supplemented with 500 mM NaCl and 1 mM dithiothreitol using a Chirascan spectropolarimeter (Applied Photophysics) and a cuvette of 0.1 cm path length. Thermal unfolding profiles between 5 °C and 80 °C were recorded by increasing the temperature at a ramping rate of 1 °C min$^{-1}$ monitoring the CD signal at 222 nm. Midpoints of thermal unfolding profiles were determined using the Global3 program (Applied Photophysics).

**Structural analysis of crystallographic NcKar9N homodimer interfaces**
The three crystallographic NcKar9N homodimer interfaces were analysed using the PDBePISA server[49] on Protein Data Bank (PDB) ID 7AG9. The following residues form the three different interfaces: α, F181, Q184, E185, F188, F288, E292, E299, and K303, and Y318′, E322′, K329′, R343′, F344′, Q347′, K350′ and K351′; β, L107, E111, D123, F124, L127 and I131, and K179′, R233′, K234′, L236′, I237′ and R240′; γ, N360, Y363, R364, E367, R390 and L398, and N211′, N212′, Y363′, R364′ and L377′. The prime discriminates the second protomer of a respective crystallographic dimer. Figures were prepared using the software PyMOL (The PyMOL Molecular Graphics System, Schrödinger, LLC).

**Yeast strains and cloning**
Yeast strains used in this study were created by transformation of PCR-amplified fragments that integrated into the genome by homologous recombination or crossing, sporulation and tetrad dissection. Notably, for the creation of Kar9 interface mutations, mutant primers were used to PCR amplify the *KAR9* open reading frame (ORF) as multiple fragments overlapping at the sites of the point mutations. Co-transformation of the mutant fragments into a *kar9Δ* strain yielded the full mutant ORF[4]. N-terminal mNeonGreen tags on Bim1 and Bik1 were introduced by transformation of the wild-type allele with PCR-amplified fragments with homology up- and downstream of the start codon to insert the tag and a selectable marker flanked by loxP sites. Expression of Cre recombinase from a transiently introduced plasmid was used to excise the marker after selection. For expression of a second genomic copy of Dam1- or Kar9-mNeonGreen, these ORFs were first cloned into an integrative plasmid, which was then linearized and integrated at the *URA3* locus. Transformants were verified by PCR and sequencing, and spores by replication onto relevant selection plates. All yeast strains and primers are summarized in Supplementary Table 1.

**Yeast viability assay**
Haploid yeast strains of the given genotypes were spotted in 10× serial dilutions, starting from OD$_{600}$ 0.05, by spotting 4 µl from the individual dilutions on YPD plates and incubating them at the indicated temperatures for 2–3 days.

**Yeast live-cell time-lapse fluorescent microscopy**
For live-cell time-lapse fluorescent microscopy, yeast strains were exponentially grown in synthetic medium lacking tryptophan at 25 °C, collected by centrifugation at 600g for 1.5 min, and placed on glass slides for imaging at 25 °C. For the phenotypes late-anaphase spindle alignment (Fig. 4c), spindle accumulation (Fig. 4d), Kar9 fluorescence intensity (Fig. 5a) and Kar9 asymmetry (Fig. 5b), cells were imaged on a Personal DeltaVision microscope (Cytiva) at 100× magnification with 2 × 2 binning (pixel size 128 nm) in CFP and FITC channels (*kar9Δ* only CFP) over ten iterations separated by 13 s in 17 z-slices separated by 0.3 µm. For microtubule plus-end tracking (Figs. 5c–g and 7a,b), cells were imaged on a Visitron Spinning Disk microscope used in wide field mode at 150× magnification (100× objective with 1.5× auxiliary magnification) and 2 × 2 binning (pixel size 173 nm) with simultaneous acquisition in mCherry and GFP channels over 50 iterations separated by 3.5 s in 17 z-slices separated by 0.3 µm.

For hexanediol/nocodazole treatment, cells were grown on YPD agar plates at low density and then resuspended in synthetic medium lacking tryptophan supplemented with 10 µg ml$^{-1}$ digitonin and with 5% (w/v) 1,6-hexanediol, 30 µg ml$^{-1}$ nocodazole or none of them. Imaging was performed as described above with a DeltaVision Personal microscope.

**Microscopy data analysis**
For late-anaphase spindle alignment and spindle accumulation, the seven most in-focus planes of the z-stack were sum projected and the phenotypes were manually classified in the second timeframe. The other timeframes were used to assign cells to the analysed cell cycle stage. Cells were considered to be in late anaphase if the spindle reached at least the approximate length of the mother compartment.

For Kar9-mNeonGreen fluorescence intensity quantification, the complete z-stacks of timeframes 2–10 were maximum projected in Fiji[50] and the background (slide)-subtracted maximum intensity for each metaphase cell was measured and averaged over the nine timeframes. For mNeonGreen-Bim1 and -Bik1, the fluorescence at SPBs or within the spindle was excluded. For Kar9 asymmetry and cumulative fluorescence intensity, background (cell)-subtracted intensities (integrated densities) were measured on both sides of the spindle in timeframe 2. Cumulative fluorescence intensity was calculated as the sum of intensities on both sides. The Kar9 asymmetry index was calculated as the absolute difference divided by the sum of intensities on opposite ends of the spindle $\left( \frac{|I_{bud} - I_{mother}|}{I_{bud} + I_{mother}} \right)$.

For microtubule plus-end tracking analysis, the complete z-stacks of all 50 timeframes were maximum projected. Kar9 punctum loss (Fig. 5e) was classified manually for these movies. Further, for each Kar9 punctum the associated SPB was determined by looking at

the trajectory of Kar9 fluorescence. The 2D distance between the Kar9-mNeonGreen punctum and the relevant Spc42-mCherry focus (microtubule length) as well as the background-subtracted (cell background) integrated fluorescence intensity was measured for each timeframe. Smoothed (moving average over three timeframes) microtubule length trajectories are shown in Fig. 5d.

Periods of microtubule growth and shrinkage were annotated automatically on non-smoothed trajectories. For each Kar9 punctum, the procedure involved: (1) applying a Savitzky–Golay finite impulse response smoothing filter (polynomial order 1, frame length 5) to the microtubule length trajectory; (2) identifying change points in the filtered data (maximum of five change points per trajectory, comparison with least squares linear fit, minimal distance between change points of five frames); (3) detecting minima/maxima in the unfiltered trajectories closest to the identified change points (minimum required peak prominence of 25% of the difference between maximal and minimal microtubule length); and (4) annotating periods as growth (shrinkage) when slopes of linear regressions on the partial trajectories between maxima/minima (or change points) were significantly positive (negative) based on $t$-tests on coefficients of linear regressions, with $\alpha = 0.05$. To estimate initial microtubule length or intensity as well as rate of microtubule length and fluorescence change separately for individual microtubule growth or shrinkage phases and strains, we used ordinary linear regression on the respective partial trajectories. For joint estimation (over length or intensity data for all strains during growth or shrinkage; Fig. 5f,g), we used linear mixed effects models[51] where the interactions between intercept and slope with strain were the fixed effects, and intercept and slope could vary by partial trajectories. Model parameters were estimated by maximum likelihood estimation and significance of fixed-effects coefficients was determined by two-sided $t$-tests. Automatic classification and regression were implemented in MATLAB R2019b.

For stoichiometry analysis, the $z$-stacks were maximum projected and Kar9 and Dam1 intensities (raw integrated density) were measured after subtracting the cell background. For estimating the brilliance of one Dam1-mNeonGreen molecule, we divided the median value of metaphase punctum intensity (912) by 272 (16 kinetochores × 17 molecules per kinetochore). We assumed that the brilliance of one Dam1-mNeonGreen of 3.35 a.u. is similar to that of one molecule of Kar9-mNeonGreen. Thus, to calculate the amount of Kar9 molecules at the microtubule tip during metaphase we divided the mean value of the Kar9 punctum intensity during metaphase (244 a.u.) by the calculated brilliance of one mNeonGreen molecule (3.35 a.u.), resulting in 72.9 Kar9 molecules at the tip of one astral microtubule.

For calculating the levels of overexpressed Kar9, we multiplied the measured intensity of the punctum by the factor (2.5) between the exposure times of overexpressed (100 ms) and endogenous Kar9 or two copies of *KAR9* (250 ms).

For significance analysis, we used two-tailed Welch's $t$-test for continuous data and two-tailed two-proportion $z$-test for nominal data. The used test is indicated in the figure legend for each panel.

### Disorder prediction
For prediction of disordered protein regions, flDPnn[52] was used.

### Statistics and reproducibility
No statistical method was used to pre-determine sample size. No data were excluded from the analyses. The experiments were not randomized. The investigators were not blinded for data analysis. Owing to high protein consumption, phase separation and other in vitro experiments (Figs. 1d,e, 2b,d,e, 3a and 7c and Extended Data Figs. 1 and 2d,e) were performed only once. For live cell experiments, the number of biological replicates (genetically identical clones) are indicated in the figures or figure legends. Notably, tip tracking experiments

(Figs. 5e,f,g and 7b and Extended Data Fig. 8), stoichiometry analysis (Fig. 6) and hexanediol experiments (Fig. 7d,e and Extended Data Fig. 9) were performed only once.

For a summary of materials and software used in this manuscript, see Supplementary Table 1.

### Reporting summary
Further information on research design is available in the Nature Portfolio Reporting Summary linked to this article.

## Data availability
All data and any additional information required to re-analyse the data reported in this paper are available from the lead contact upon request. Further information and requests for resources and reagents should be directed to and will be fulfilled by the lead contact, Y.B. (yves.barral@bc.biol.ethz.ch). The previously published and re-analysed structure of *Naumovozyma castellii* Kar9 N-terminal domain is available from PDB under accession code 7AG9. Source data are provided with this paper.

## Code availability
Code used for data analysis is available from the lead contact upon request.

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

## Acknowledgements
We thank M. Choudhury, M. Bayer and T. Vukovic for their help with the experiments, M. Stangier for helping with protein production and ScopeM for their support and assistance with microscopy. We thank previous and current members of the Steinmetz, Dufresne and Barral laboratories for extensive and critical discussions throughout the project. This work was supported by a Marie Curie COFUND fellowship (to A.K.) and by grants from the Swiss National Science Foundation (31003A-105904 to Y.B. and 31003A_166608 and 310030_192566 to M.O.S.; Sinergia CRSII5_189940 to Y.B., M.O.S. and E.R.D.) and from SystemsX.ch (RTD grant #2012/192 TubeX, to M.O.S., J.S. and Y.B.).

## Author contributions
The authors S.M.M., A.-M.F. and A.K. contributed equally. S.M.M., A.-M.F. and A.K. designed and performed the research, and analysed the data. M.I. characterized the material properties of the different

droplets in vitro. J.S. designed and executed the quantitative analysis of Kar9 levels on shrinking microtubules. R.T.B. designed and performed the analysis of Kar9 overexpression with A.-M.F. E.R.D. contributed to designing the research and supervised the work of M.I. Y.B. and M.O.S. designed and supervised the research, analysed the data and wrote the manuscript with input from all authors.

## Competing interests

The authors declare no competing interests.

## Additional information

**Extended data** is available for this paper at https://doi.org/10.1038/s41556-022-01035-2.

**Correspondence and requests for materials** should be addressed to Michel O. Steinmetz or Yves Barral.

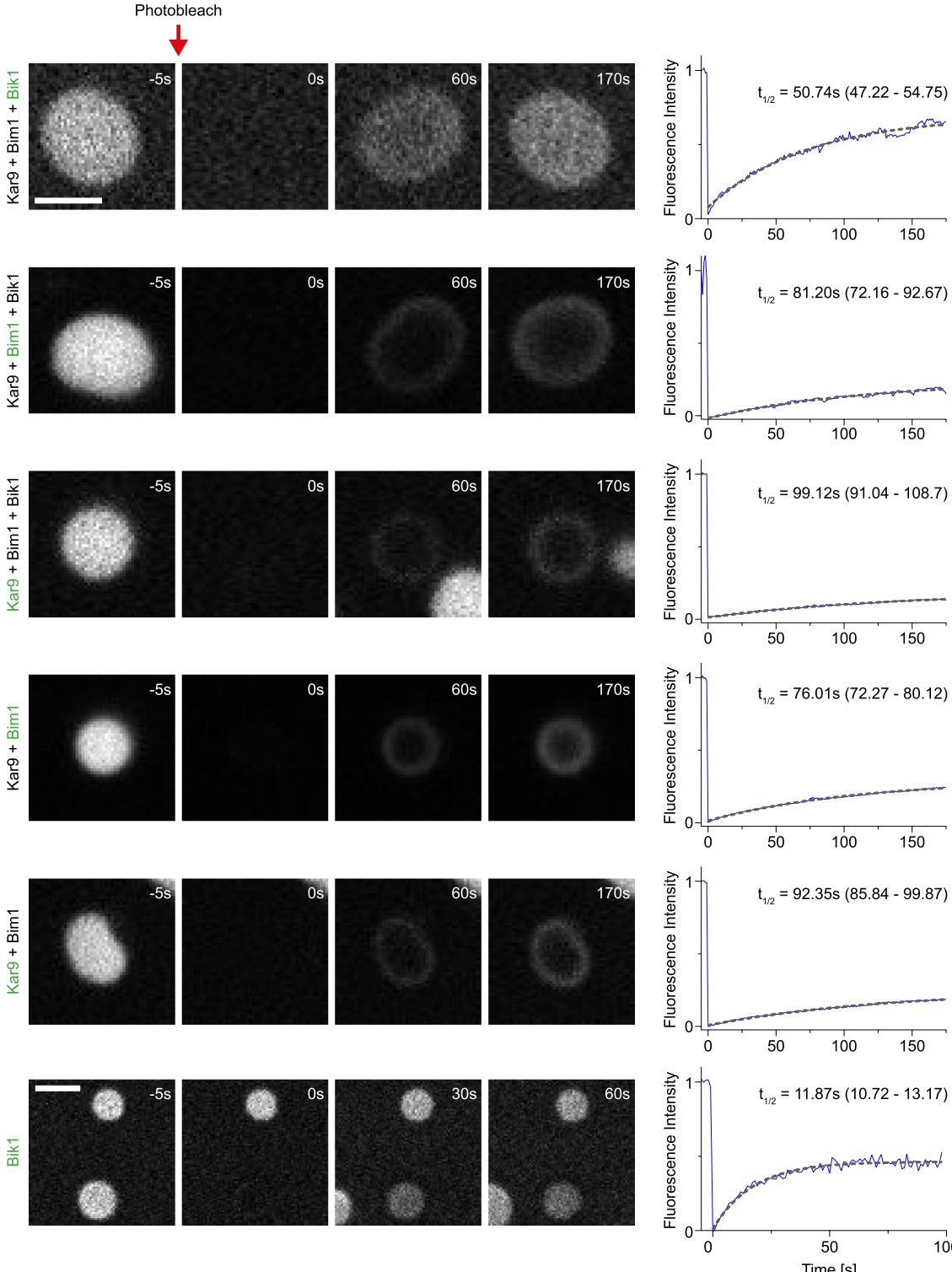

**Extended Data Fig. 1 | Fluorescence Recovery after Photobleaching (FRAP) of in vitro protein droplets.** Time series (left) displaying fluorescence recovery in the indicated protein droplets after a bleaching laser pulse and quantification (right). Tenfold less concentrated labeled protein (mNeonGreen-Bik1, Bim1-GFP, or Kar9-GFP, indicated with green letter fonts) was added to mixtures of unlabeled protein (5 μM each). Blue line displays fluorescence intensity over time of a single experiment normalized to prebleached droplet intensity, grey dotted line is a single exponential decay fitted to the recovery period. Half-lives ($t_{1/2}$) of the fitted decay are given with 95% confidence intervals of the fit. For Bik1 droplets, the quantified intensity was corrected for the effect of defocusing according to unbleached reference droplets. Scale bar 3 μm. Single experiment. Related to Fig. 1. Source numerical data available in source data.

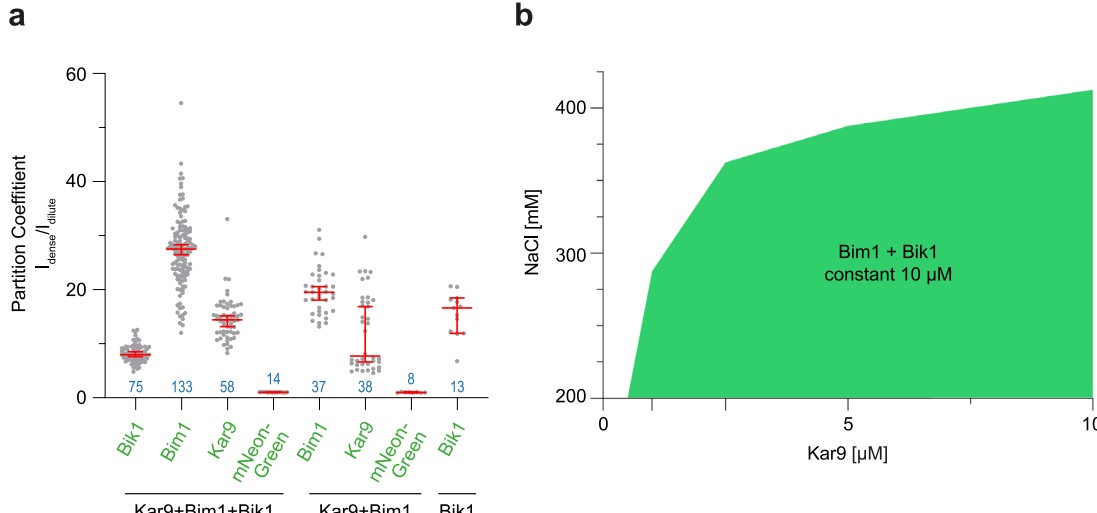

**Extended Data Fig. 2 | Partition coefficients and phase diagram with variable Kar9 in constant Bim1 and Bik1. a**, Partition coefficients (ratio between fluorescence intensity inside and outside of droplets) for mNeonGreen-Bik1, Bim1-GFP, Kar9-GFP, and mNeonGreen into non-fluorescent, equimolar mixtures of Kar9 + Bim1 + Bik1, Kar9 + Bim1, and Bik1 droplets (10 μM each;

median with 95% confidence interval). Blue values indicate *n* analyzed droplets, single experiment. **b**, Phase diagram of Kar9 that was titrated into an equimolar mixture of Bim1 + Bik1 (10 μM each). Single experiment. Related to Fig. 1. Source numerical data available in source data.

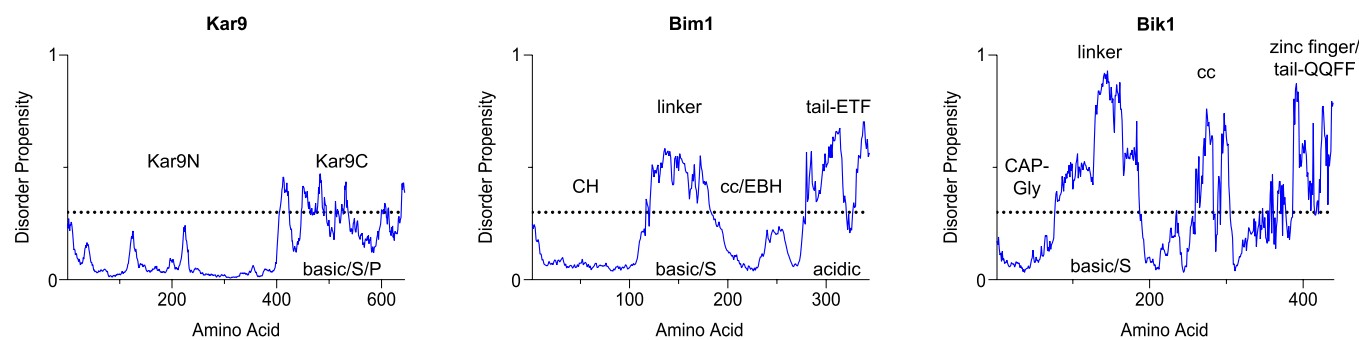

**Extended Data Fig. 3 | Disorder prediction for Kar9, Bim1, and Bik1.** Disorder propensity calculated by flDPnn[52] visualizing the disordered regions of Kar9, Bim1, and Bik1 with annotations for known folded domains and amino acid compositional biases. The 0.3 threshold level for disorder is indicated by dotted lines. Related to Fig. 2. Source numerical data available in source data.

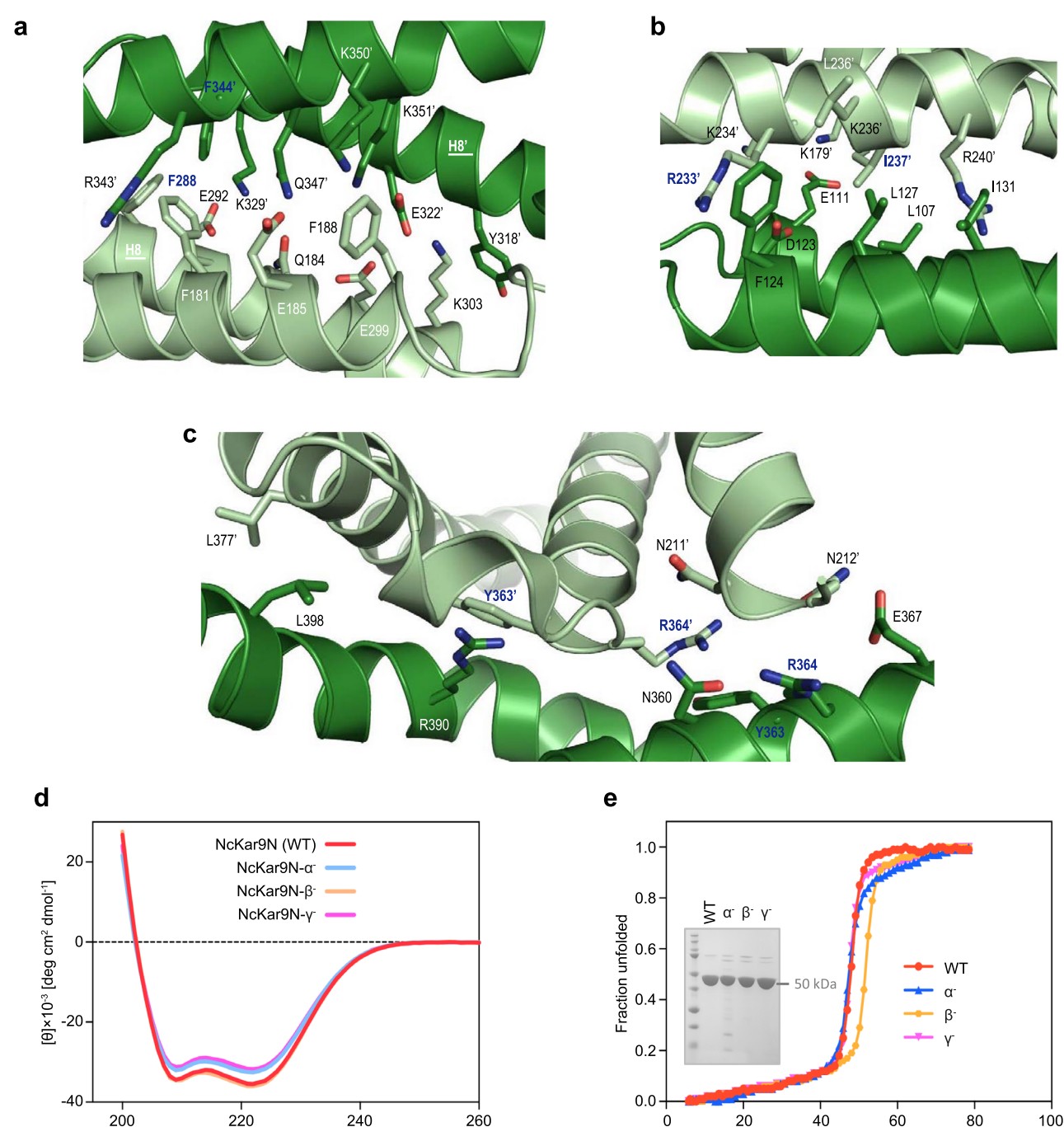

**Extended Data Fig. 4 | Protomer contacts forming the NcKar9N crystal and NcKar9 dimer interface mutagenesis. a-c**, Close-up views of the α **(a)**, β **(b)**, and γ **(c)** interfaces seen between protomers in the NcKar9N crystal (PDB ID 7AG9). Interacting residue side chains are shown in stick representation. Oxygen and nitrogen atoms are colored in red and blue, respectively; carbon atoms are in dark green for chain A and in light green for chains B, C, and D. Interface residues that were mutated in the three crystallographic dimers are labeled in bold and blue lettering. **d** and **e**, Far-ultraviolet CD spectra **(d)** and thermal denaturation profiles **(e)** of His-NcKar9N (red), His-NcKar9N-α⁻ (blue), His-NcKar9N-β⁻ (yellow), and His-NcKar9N-γ⁻ (magenta). Inset in **(e)**, SDS-PAGE analysis of the His-NcKar9N variants. Spectra are averages of 3 measurements of the same sample, denaturation profiles are single experiments. Related to Fig. 2.

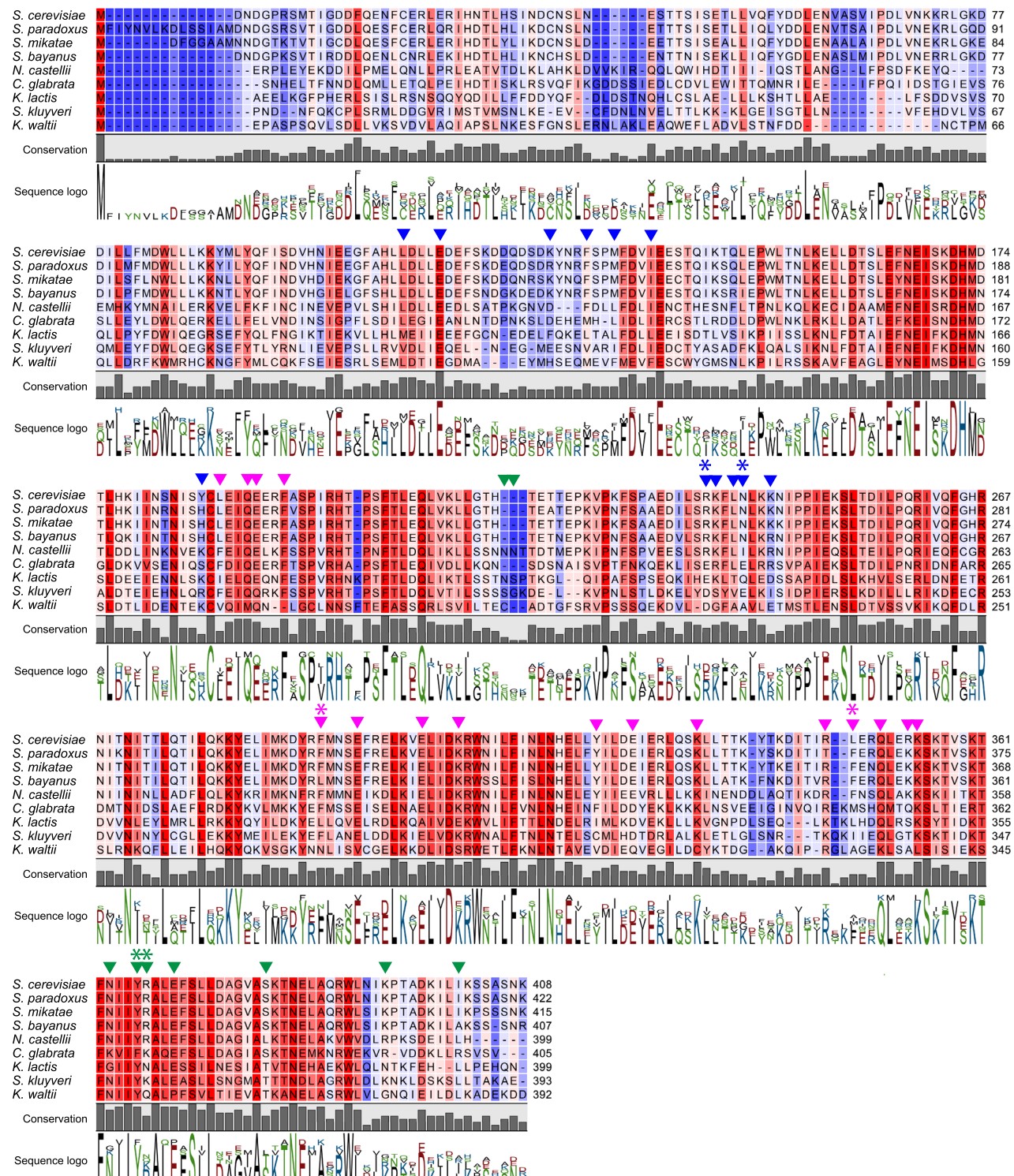

**Extended Data Fig. 5 | Multiple sequence alignment of Kar9 homologues.**
Multiple sequence alignment of Kar9 homologues of closely and distantly related yeast species. Conserved amino acid residues are indicated by the background color from strong (red) to no (blue) conservation, as well as by grey bars below the alignment. Consensus amino acids are indicated by the size of the letters below the alignment. Kar9 self-oligomerization interfaces are denoted by magenta (α), blue (β) and green (γ) arrowheads. Interface residues that were mutated in this study are indicated by asterisk in identical color. Related to Fig. 2.

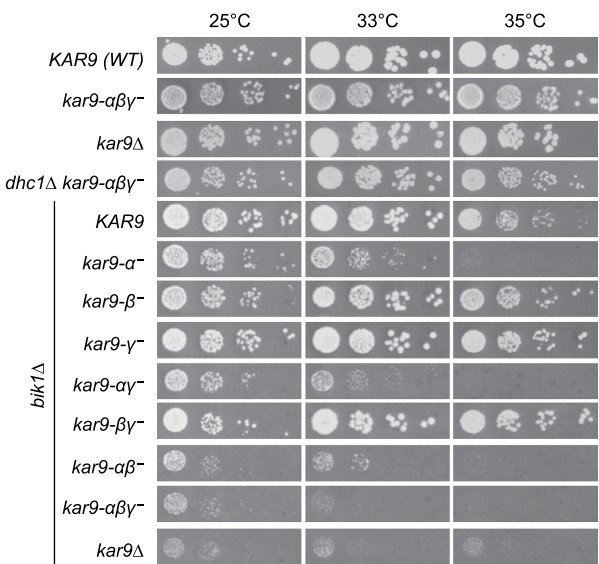

**Extended Data Fig. 6 | Budding yeast growth spot assay.** Spot assays with serial 10X dilutions starting at OD600 = 0,05 performed at 25, 33, and 35 °C. Single experiment. Related to Fig. 4a.

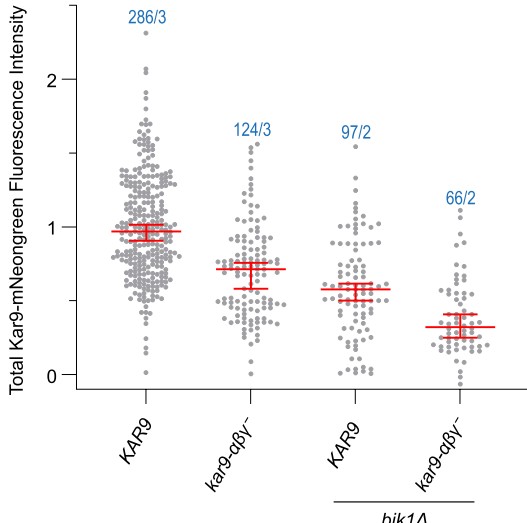

**Extended Data Fig. 7 | Kar9 quantification on astral microtubules.** Quantification of cumulative Kar9 fluorescence intensity on astral microtubules emanating from both spindle poles (normalized median with 95% confidence interval). Unless specified in the legend. Blue values indicate *n* = cells analyzed / number of biological replicates. Related to Fig. 5a,b. Source numerical data available in source data.

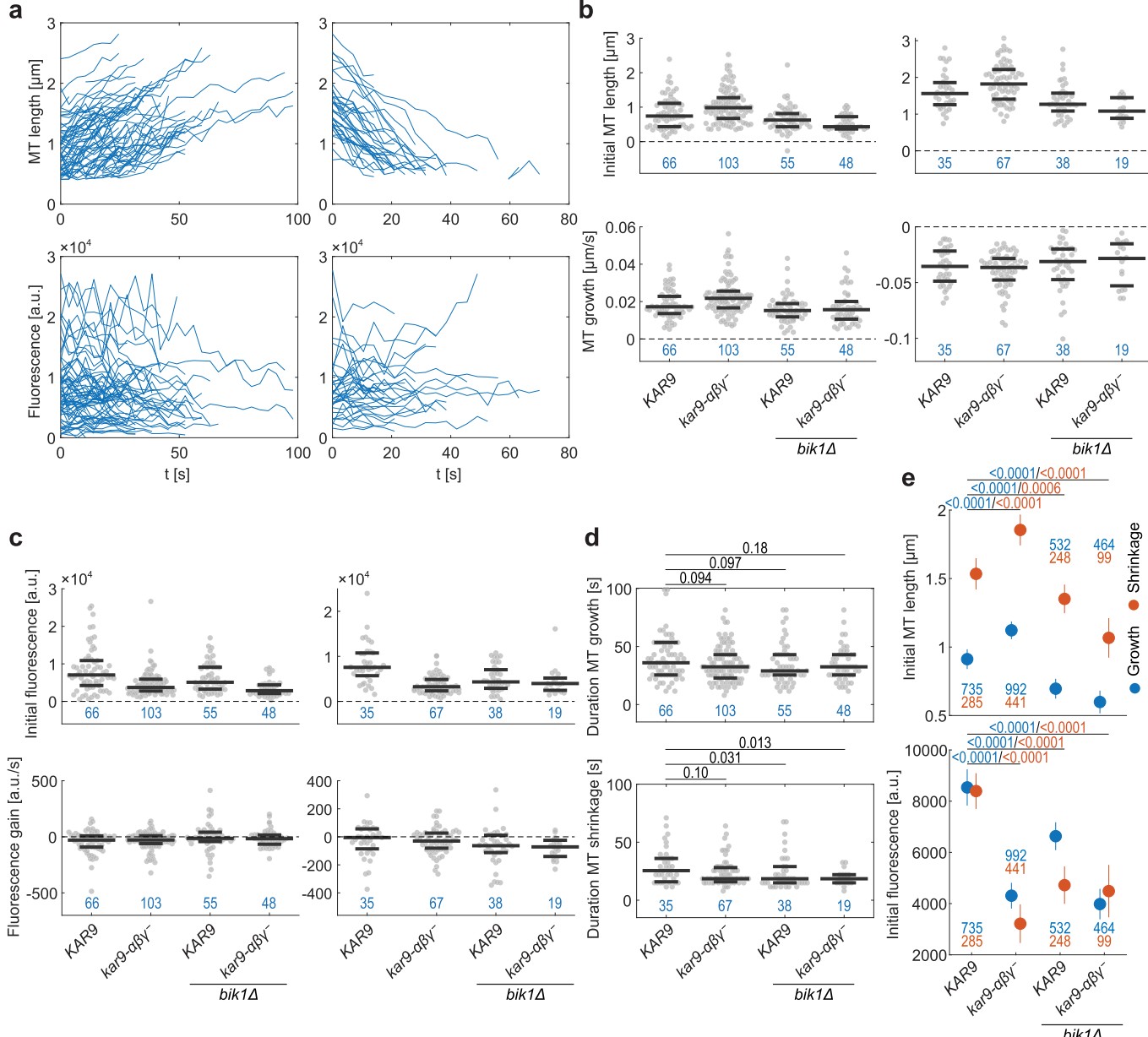

**Extended Data Fig. 8 | Analysis of microtubule and Kar9 puncta dynamics.**
**a**, Trajectories of microtubule (MT) length (top) and Kar9-mNeonGreen fluorescence intensity (bottom) extracted during individual microtubule growth (left) and shrinkage (right) phases in wild-type cells. **b**, Estimates of initial microtubule length (top; defined as the length at the start of a growth or shrinkage phase) and growth/shrinkage rate (bottom) during microtubule growth (left) and shrinkage (right) phases using linear regression for individual trajectories in the indicated strains. Dots, estimates for individual trajectories; long lines, medians; short lines, 50% quantiles. **c**, As panel **b** but for fluorescence intensity of Kar9 puncta. **d**, Durations of microtubule growth and shrinkage

phases. P values were determined by Wilcoxon rank sum test. For panels **b-d**, blue values indicate n = number of microtubule growth/shrinkage events analyzed, single experiment. **e**, As Fig. 5f,g, but showing the linear mixed-effects (LME) model estimates of initial microtubule lengths and Kar9-mNeonGreen fluorescence intensity levels at the start of microtubule growth and shrinkage phases. Color of P values (two-sided t-test) and n (number of timeframes considered for the model, single experiment) matches the growth/shrinkage category they refer to. Related to Fig. 5. Source numerical data available in source data.

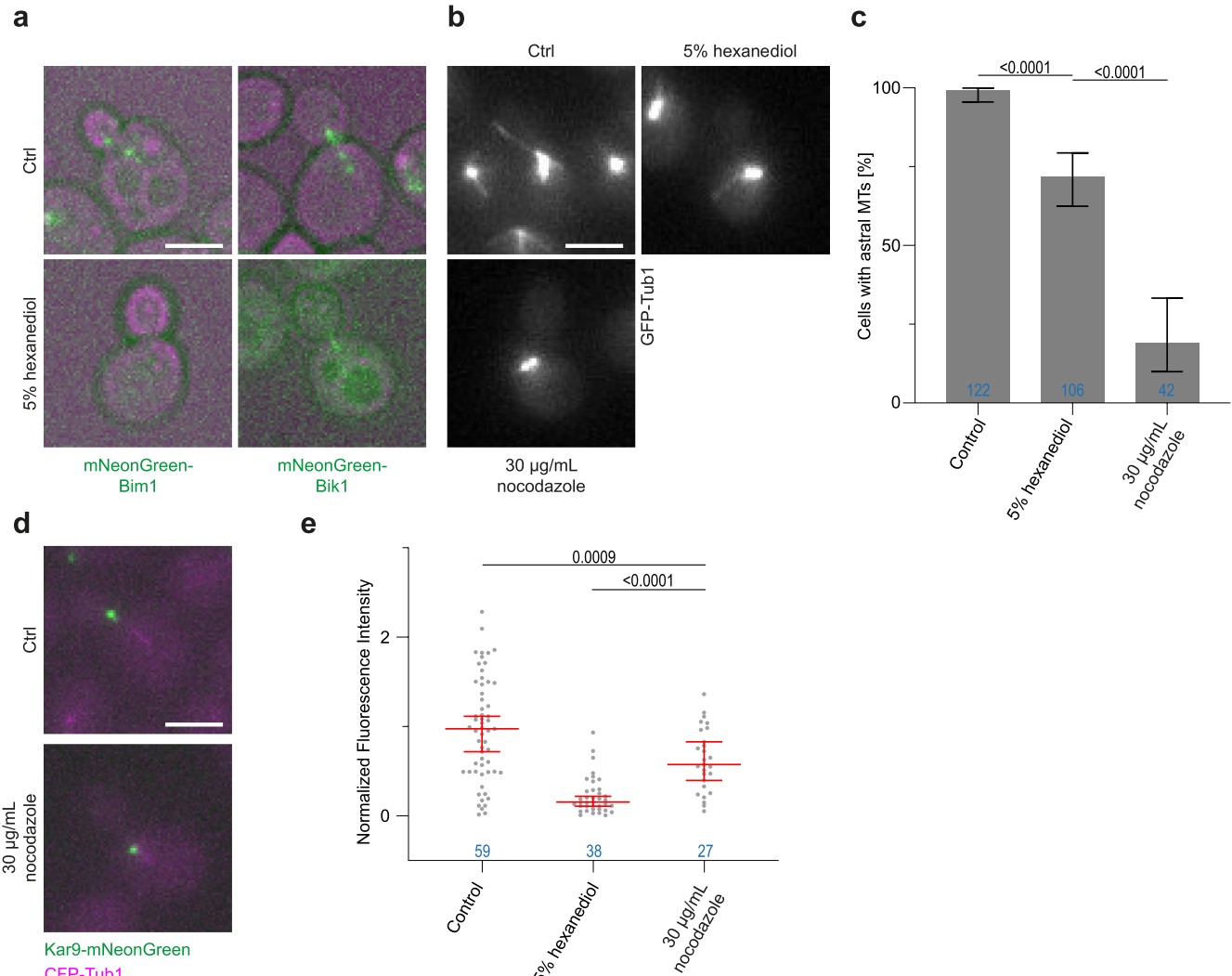

**Extended Data Fig. 9 | Hexanediol and nocodazole treatment. a**, Micrographs of mNeonGreen-Bim1 and mNeonGreen-Bik1 expressing control and 5% hexanediol treated cells as quantified in Fig. 7e. Since these cells did not express CFP-Tub1, the transmission channel is used as a reference in magenta. Scale bar, 3 µm. **b**, Micrographs of GFP-Tub1 expressing control, 5% hexanediol, and 30 µg/mL nocodazole treated cells. **c**, Frequency of cells with visible astral microtubules labeled with GFP-Tub1 in control, 5% hexanediol, and 30 µg/mL nocodazole treatment (error bars are Wilson/Brown 95% confidence intervals determined from binomial distribution). *P* values above bars are determined by two-proportion z test. **d**, Micrographs of cells expressing Kar9-mNeonGreen treated with 30 µg/mL nocodazole versus control. **e**, Kar9-mNeonGreen fluorescence intensity levels in wild-type cells treated with 5% hexanediol or 30 µg/mL nocodazole versus untreated control (normalized median with 95% confidence intervals, the control and 5% hexanediol data are the same as shown in Fig. 7e). *P* values were determined by Welch's *t*-test. Blue values in panels **c,e** indicate *n* = number of cells analyzed, single experiment. Related to Fig. 7d,e. Source numerical data available in source data.

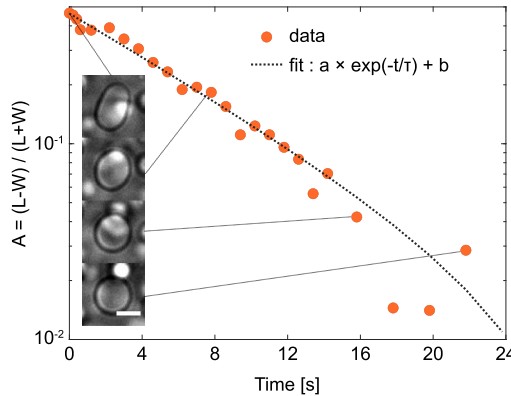

**Extended Data Fig. 10 | Droplet fusion time quantification example.** Fusion of two Bik1 droplets at a later stage can be considered as a uni-axially deformed droplet reaching its final spherical state, driven by surface tension. Parameter A relates the major and minor axis of the droplet (L and W, respectively) and is plotted versus time, where an exponential fit allows to calculate the fusion time τ. Example of the analysis of a single fusion event. Scale bar in the inlet, 5 μm. Related to Fig. 1e.

# Reporting Summary

## Statistics

For all statistical analyses, confirm that the following items are present in the figure legend, table legend, main text, or Methods section.

| n/a | Confirmed | |
|---|---|---|
| ☐ | ☒ | The exact sample size (*n*) for each experimental group/condition, given as a discrete number and unit of measurement |
| ☐ | ☒ | A statement on whether measurements were taken from distinct samples or whether the same sample was measured repeatedly |
| ☐ | ☒ | The statistical test(s) used AND whether they are one- or two-sided <br> *Only common tests should be described solely by name; describe more complex techniques in the Methods section.* |
| ☒ | ☐ | A description of all covariates tested |
| ☐ | ☒ | A description of any assumptions or corrections, such as tests of normality and adjustment for multiple comparisons |
| ☐ | ☒ | A full description of the statistical parameters including central tendency (e.g. means) or other basic estimates (e.g. regression coefficient) AND variation (e.g. standard deviation) or associated estimates of uncertainty (e.g. confidence intervals) |
| ☐ | ☒ | For null hypothesis testing, the test statistic (e.g. *F*, *t*, *r*) with confidence intervals, effect sizes, degrees of freedom and *P* value noted <br> *Give P values as exact values whenever suitable.* |
| ☒ | ☐ | For Bayesian analysis, information on the choice of priors and Markov chain Monte Carlo settings |
| ☒ | ☐ | For hierarchical and complex designs, identification of the appropriate level for tests and full reporting of outcomes |
| ☒ | ☐ | Estimates of effect sizes (e.g. Cohen's *d*, Pearson's *r*), indicating how they were calculated |

*Our web collection on statistics for biologists contains articles on many of the points above.*

## Software and code

Policy information about availability of computer code

| Data collection | For data collection, the commercial software of the microscope manufacturers was used. Specifically, softWoRx 6.5.2 was used for collection on Personal DeltaVision, VisiView 5 was used for collection on Visitron Spinning Disk. ASTRA 6 was used to acquire elution profiles during MALS erperiments. |
|---|---|
| Data analysis | For data analysis, Fiji 2.3.0/ImageJ 1.53 was used for image analysis, Excel 16 and Matlab R2019b were used for calculations and statistics, Graphpad Prism 9 was used for plots and statistics, ASTRA 6 was used to estimate molecular weights in MALS experiments, Global3 was used to interpret CD spectra, Pymol 2.5 was used for to create structure representations, CLC Genomics Workbench was used to create sequence alignments. |

For manuscripts utilizing custom algorithms or software that are central to the research but not yet described in published literature, software must be made available to editors and reviewers. We strongly encourage code deposition in a community repository (e.g. GitHub). See the Nature Portfolio guidelines for submitting code & software for further information.

## Data

Policy information about availability of data

All manuscripts must include a data availability statement. This statement should provide the following information, where applicable:
- Accession codes, unique identifiers, or web links for publicly available datasets
- A description of any restrictions on data availability
- For clinical datasets or third party data, please ensure that the statement adheres to our policy

All data and any additional information required to analyze the data reported in this paper are available from the lead contact upon request. The previously published crystal structure of Naumovozyma Castellii Kar9 N-terminal domain is available under PDB ID 7AG9.

# Field-specific reporting

Please select the one below that is the best fit for your research. If you are not sure, read the appropriate sections before making your selection.

☒ Life sciences    ☐ Behavioural & social sciences    ☐ Ecological, evolutionary & environmental sciences

For a reference copy of the document with all sections, see nature.com/documents/nr-reporting-summary-flat.pdf

# Life sciences study design

All studies must disclose on these points even when the disclosure is negative.

| | |
|---|---|
| Sample size | No statistical method was used to predetermine sample size. Sample size (cell number) was chosen based on effect size and variance within the sample. We systematically analyzed all cells in the correct cell cycle stage at least until statistical significance was reached for clear phenotypes with importance to the manuscript. For most experiments, the observed differences in phenotypes were large. Therefore even 50 cells were often more than enough. |
| Data exclusions | No data were excluded from the analyses. |
| Replication | For most live-cell microscopy data, biological duplicates or triplicates were used and images were acquired on multiple days (technical replicates). The only exception is the tip tracking assay, which is very demanding in terms of data analysis. There, only 1 clone per strain was analyzed. Due to high protein consumption, phase separation experiments were not systematically performed with replication (replication showed no significant differences where it was performed). |
| Randomization | The experiments were not randomized. In this study, the phenotypes of pure clones with known engineered genotypes were compared. Thus, no experiments required randomization to eliminate confounding effects. |
| Blinding | The investigators were not blinded for data analysis. The mutant strains were often discernible by eye compared to the wild type control, due to the very obvious differences in the phenotypes. Thus, blinding was not truly feasible. |

# Reporting for specific materials, systems and methods

We require information from authors about some types of materials, experimental systems and methods used in many studies. Here, indicate whether each material, system or method listed is relevant to your study. If you are not sure if a list item applies to your research, read the appropriate section before selecting a response.

## Materials & experimental systems

| n/a | Involved in the study |
|---|---|
| ☒ ☐ | Antibodies |
| ☒ ☐ | Eukaryotic cell lines |
| ☒ ☐ | Palaeontology and archaeology |
| ☒ ☐ | Animals and other organisms |
| ☒ ☐ | Human research participants |
| ☒ ☐ | Clinical data |
| ☒ ☐ | Dual use research of concern |

## Methods

| n/a | Involved in the study |
|---|---|
| ☒ ☐ | ChIP-seq |
| ☒ ☐ | Flow cytometry |
| ☒ ☐ | MRI-based neuroimaging |

