## [Peer Review File · Nature Cell Biology]

Peer Review Information

Journal: Nature Cell Biology

Manuscript Title: Multivalency ensures persistence of a +TIP-body at specialized microtubule ends

Corresponding author name(s): Professor Yves Barral

Editorial Notes:

Reviewer Comments & Decisions:

Decision Letter, initial version:

*Please delete the link to your author homepage if you wish to forward this email to co-authors.

Dear Professor Barral,

I hope you are well and I apologize for the very long delay in the peer review of your submission.

Your manuscript, "High interaction valency ensures cohesion and persistence of a microtubule +TIP body at the plus end of a single specialized microtubule in yeast", has now been seen by 3 referees, who are experts in microtubules (referee 1); biomolecular condensation (referee 2); and spindle microtubules (referee 3). As you will see from their comments (attached below) they find this work of potential interest, but have raised substantial concerns, which in our view would need to be addressed with considerable revisions before we can consider publication in Nature Cell Biology.

Nature Cell Biology editors discuss the referee reports in detail within the editorial team, including the chief editor, to identify key referee points that should be addressed with priority, and requests that are overruled as being beyond the scope of the current study. To guide the scope of the revisions, I have listed these points below. We are committed to providing a fair and constructive peer-review process, so please feel free to contact me if you would like to discuss any of the referee comments further.

I should stress that the referees' concerns point to unclear assessment of Kar9 subcellular localization and behaviour, as well as unclear assessment of biomolecular condensation which would need to be addressed with experiments and data, and reconsideration of the study for this journal and re-engagement of referees would depend on strength of these revisions.

In particular, it would be essential to:

A) Assess Kar9 subcellular localization (Reviewers #1 and #2)

B) Completely characterize the biomolecular condensate behaviour of, and effects of, Kar9 and Bim1 and Bik1 (all Reviewers)

C) Investigate the regulatory relationship between Kar9, Bik1, and Bim1 behaviour on microtubule spindles (all Reviewers)

D) All other referee concerns pertaining to strengthening existing data, providing controls, methodological details, clarifications and textual changes, should also be addressed.

E) Finally please pay close attention to our guidelines on statistical and methodological reporting (listed below) as failure to do so may delay the reconsideration of the revised manuscript. In particular please provide:

We would be happy to consider a revised manuscript that would satisfactorily address these points, unless a similar paper is published elsewhere, or is accepted for publication in Nature Cell Biology in the meantime.

- ensure that it conforms to our format instructions and publication policies (see below and <https://www.nature.com/nature/for-authors>).
- provide a point-by-point rebuttal to the full referee reports verbatim, as provided at the end of this letter.
- provide the completed Reporting Summary (found here <https://www.nature.com/documents/nr-reporting-summary.pdf>). This is essential for reconsideration of the manuscript will be available to editors and referees in the event of peer review. For more information see <http://www.nature.com/authors/policies/availability.html> or contact me.

When submitting the revised version of your manuscript, please pay close attention to our [href="https://www.nature.com/nature-research/editorial-policies/image-integrity">Digital Image Integrity Guidelines](https://www.nature.com/nature-research/editorial-policies/image-integrity). and to the following points below:

Nature Cell Biology is committed to improving transparency in authorship. As part of our efforts in this direction, we are now requesting that all authors identified as 'corresponding author' on published papers create and link their Open Researcher and Contributor Identifier (ORCID) with their account on the Manuscript Tracking System (MTS), prior to acceptance. ORCID helps the scientific community achieve unambiguous attribution of all scholarly contributions. You can create and link your ORCID from the home page of the MTS by clicking on 'Modify my Springer Nature account'. For more information please visit www.springernature.com/orcid.

This journal strongly supports public availability of data. Please place the data used in your paper into a public data repository, or alternatively, present the data as Supplementary Information. If data can only be shared on request, please explain why in your Data Availability Statement, and also in the correspondence with your editor. Please note that for some data types, deposition in a public repository is mandatory - more information on our data deposition policies and available repositories appears below.

[Redacted]

We would like to receive a revised submission within six months.

We hope that you will find our referees' comments, and editorial guidance helpful. Please do not hesitate to contact me if there is anything you would like to discuss.

Best wishes,

Daryl Jason David

Daryl J.V. David, PhD

Senior Editor, Nature Cell Biology
Consulting Editor, Nature Communications
Nature Portfolio

Heidelberger Platz 3, 14197 Berlin, Germany
Email: daryl.david@nature.com
ORCID: <https://orcid.org/0000-0002-9253-4805>

Reviewers' Comments:

Reviewer #1:

Remarks to the Author:

In their manuscript, Meier et al. discover a network of proteins that form what they coin the "+TIP body", and show that the underlying mechanisms that lead to the formation of this body is liquid-liquid phase separation.

Microtubules are evolutionary conserved cytoskeletal found in all eukaryotes. They interact with a multitude of different proteins of which more and more were discovered over the past decades. Given this multitude of interactors, plus the fact that microtubules are highly similar within a single cell, one of the burning questions in cell biology is how single microtubules can acquire specific functions that clearly distinguish them from their neighbours. One intuitive mechanism is that they are able to acquire a unique set of interacting proteins, which in turn assure their functional specialization. The current manuscript demonstrates how low-affinity interactions between multiple proteins at the plus-tip (+TIP) of microtubules forms by liquid-liquid phase separation, and how this "Kar9-body" determines cytoplasmic microtubule functions.

Liquid-liquid phase separation has been intensely studied in the past years, however, direct functional studies in cells have remained rather scarce. The current work takes an important step toward determining the biological role liquid-liquid phase separation in living cells. Using yeast as a model system has a number of important advantages, out of which the most important might be the small number of microtubules in yeast cells. This provides the great advantage that the function of single, well-defined microtubules can be characterized, which would be immensely more difficult in larger cells, such as mammalian cells, which have thousands of microtubules. The results of the currently study, by contrast, are likely to apply to many different cells and organisms, as the proteins that form the Kar9-body are evolutionarily conserved.

The authors first characterise the phase-separation behaviour of three components of the +TIP network, Bim1, Bik1 and Kar9 using purified proteins. They demonstrate that phase separation of Bim1 and Bik1 is promoted by Kar9, which alone does not phase-separate but precipitates. Having defined the precise biochemical conditions required for the formation of a phase-separated "Kar9-

body”, the authors perform a series of site-specific mutagenesis studies to define the interaction sites in the proteins necessary for their interactions, and thus, for the formation of condensates. Next the authors use the power of yeast genetics to define the importance of their biochemically identified mechanisms *in vivo*. Mutating the amino acid residues that had been biochemically defined as interaction surfaces, the authors test which of these mutations affect the viability of the yeast cells. They find that only mutations of three interaction surfaces of Kar9 have a strong impact on cell viability, and only when Bik1 is deleted (Bik1 Δ). The most striking phenotype is spindle misalignment. Next, they demonstrate that the Kar9-Bik1 interface is important for the microtubule-specific localisation of the “Kar9-body” in the cell, and that perturbation of the interfaces lead to a partial loss of the characteristic back-tracking of this body on shrinking microtubules. This phenotype is further amplified when Bik1 is lacking. Finally, the authors demonstrate that the size of the Kar9-body changes with the amount of available protein in the cell, which indicates that it is generated by phase separation as is not the case for defined protein complexes. This is elegantly shown by comparison with the kinetochore protein Dam1. The authors further show examples of fusing Kap9 drops on microtubules, and the dissolution of the drops in hexanediol, both indications for their liquid character. The paper is clearly written, the experiments are meticulously performed and analysed, and figures are in most cases clear (apart from some points listed below). The observations reported in this manuscript are important and groundbreaking as they provide a novel model for the functioning of +TIP tracking based on liquid-liquid phase separation. +TIP tracking of a number of proteins on growing microtubules plays key roles in a number of microtubule-dependent biological processes, and one of the unanswered questions was how a highly divergent group of proteins (the +TIPs) could achieve the same biophysical behaviour. The discoveries in the current paper suggest that liquid-liquid phase separation, based on multivalent but not necessarily specific interactions could be a common theme in this process. The novelty and broad importance of these findings makes this manuscript a strong candidate for Nature Cell Biology.

Major Points:

1) In Fig. 5A, the authors quantify the intensity of the Kar9 fluorescence intensity – supposedly at the +TIPs of astral microtubules where it is normally localised? This is not clear and should be better specified in the text. In Fig. 5B they then show that the mutations lead to a re-localisation of Kar9 to the other side of the spindle where it is not supposed to localise. To show this, however, the authors use a more qualitative analyses than in panel A. Given that they are able to quantify Kar9-mNeonGreen intensities, why don't they connect the data in panel A and B to a single analysis where they show the re-distribution of Kar9 from one to two spindle poles in a quantitative manner. The readout could for instance be the fluorescence intensity ration between the two poles, instead of a rather arbitrary weakly / strongly asymmetric term currently used in panel B. A second parameter analysed could be the overall fluorescence of Kar9 on BOTH poles (as both represent +TIP localisation), and not only one pole as currently in panel A.

Minor points:

- 1) In the introduction, the authors write: “...spindle pole bodies (SPBs, equivalent of the metazoan centrosomes)”. The SPBs are functional equivalents of centrosomes, however not structurally. This should be pointed out for the readers that are outside the centrosome and cytoskeleton field.
- 2) Fig. 1A, 2D, 7E: It is well-appreciated that the authors use schematic drawings to guide the reader. However, they might want to use the same colour code for all the figures in the paper (compare Fig. 1A and 2D with Fig. 7E). They might also want to show their proteins approximately to scale: Bim1 is about half the size of Bik1 and Kar9, but on their drawings they are all the same size.
- 3) In several figures, the authors use a simplified nomenclature for their mutations. Though this is

somewhat explained in the figure legends, the authors might consider having a small legend directly within the figure where this nomenclature is explained, this makes the reading of the figure easier.

4) Fig. 4: It is very difficult to read this figure; the authors should make an effort to guide the reader. For instance: how do the data in panel A relate to the plots shown in panels C & D: is for instance what is labelled as tree spots in panel A the same as Kar9 followed by 3 spots in the plots? Moreover, the authors could use bars below the legend to show which subset of the diagram represents *bik1Δ* and *dyn1Δ* in order to make it more comparable to A. In the microscopy images, increasing contrast of the red channel plus a schematic drawing representing the phenotypes would improve the understandability of the figure.

5) The authors should decide if they want to call their discovery "Kar9-body" (used throughout the text) or "+TIP-body" as used in the title.

Reviewer #2:

Remarks to the Author:

In the article entitled "High interaction valency ensures cohesion and persistence of a microtubule +TIP body at the plus-end of a single specialized microtubule in yeast," by Meier, Farcas, and colleagues, the authors investigate the effect of protein valency on microtubule plus-end condensate formation and persistence. The authors focus specifically on Kar9, Bim1, and Bik1. In vitro reconstitution assays using Kar9, Bim1, and Bik1 demonstrate that each component contributes to condensate formation and phase behavior. The authors used X-ray crystallography to identify three conformations that enabled Kar9 dimerization, each with a unique binding interface. Mutation of individual interfaces altered the ability of Kar9 to undergo phase separation while mutations of combinations of interfaces had greater effect. Mutations in Kar9 also affect its ability to undergo phase separation with Bik1 and Bim1. In yeast in which Bik1 was deleted, combinations of Kar9 mutations resulted in misalignment of the mitotic spindle and increased cell death at increasing temperatures. Mutations in Kar9 also reduced the amount of Kar9 at microtubule plus-ends, and mutation of all three interfaces decreased Bik1 levels at microtubule plus-ends. Finally, the authors connect the valency of plus-end binding partners to microtubule dynamics and the liquid-like nature of plus-end condensates.

Overall, this reviewer found this manuscript to be well organized and the presented data to be excellent. The authors observations and investigation of specific binding surfaces of Kar9 and their analysis of Kar9 interactions provide important insights into the multivalent network of proteins at the plus-ends of microtubules. Their cellular experiments do a fine job at connecting in vitro observations with cellular function. However, there are some experimental details in the paper that are assumed by the authors but not shown in the data. For example, phase separation assays with Kar9 mutants are performed with either Bim1 and Bik1 or Bim1 alone, not Bik1 alone. Additionally, some of the text should be updated using common condensate field language. These comments will be addressed in greater detail below. If the authors can address the concerns outlined by this reviewer, this paper will be a good candidate for publication in Nature Cell Biology.

Major Comments:

1) Based on the authors data, Kar9 can self-associate into three different dimer conformations. Do the authors have any insights into the relative ratio of these interactions within their crystal lattices? Is one conformation preferred over the others? Relatedly, does one of the conformations promote network / condensate formation? Also, are these conformations kinetically trapped, or can they exchange between different conformations? Perhaps using their existing mutation data, the authors can provide insight into these questions.

2) As stated above, the data in Figure 3 isn't complete. The authors state in the concluding paragraph

of the section entitled "Role of Kar9 self-association for phase separation of the core Kar9-network components in vitro" that Bik1 and Kar9 interactions sites contribute to phase separation and that Bik1, Bim1, and Kar9 interactions are redundant in driving phase separation. However, the authors never test Kar9 and Bik1 alone. They should run a set of experiments to examine Kar9 behavior with Bik1 in Figure 3 to provide a complete picture of the behavior of this interaction network.

Minor Comments:

- 1) Throughout the manuscript, condensate language / jargon should be used. For example, instead of "solubilized in these droplets" use enriched in these droplets; instead of dots use puncta, etc.
- 2) Figure 1A can be removed from the manuscript. It doesn't add to an understanding of the proteins tested and is mostly redundant (and less informative) with Figure 2D.
- 3) In the first paragraph of the first results section, the authors state that Kar9 collapses into amorphous aggregates. The description should be changed because the biophysical properties of these droplets is not tested. These structures are consistent with rapidly maturing, gel-like condensates that have been frequently observed (See PMID 33148640).
- 4) Why must Bik1 contain three homodimerization domains (middle of page 6)?
- 5) Figure 2E (middle of page 6) should be Figure 2D.
- 6) In Figure 2D, please include details of domains of Kar9, Bim1, and Bik1 in the figure.
- 7) The layout of Figure 3A is a bit confusing. It would be helpful if both WT experiments were represented by a dotted line and both mutant experiments were filled in as they currently are. By having 1 WT experiment filled and one dotted, it took a bit of consideration to understand what the authors were showing the reader.
- 8) At the top of page 8, the authors comment that alpha and beta mutants reduce Kar9 levels at microtubule plus-ends by 40% and 20%, respectively. In reviewing their data, it seems that these reductions have minimal impact on cellular function, which strongly suggests that it is the formation of a Kar9 condensate that is important rather than the amount of Kar9 at the plus-end. This is consistent with other observations that correlate phase separation of molecules to cellular function rather than the discrete number of molecules (See PMID 27056844). It would be good for the authors to emphasize this point because it is quite possibly general principle of different phase separating systems.
- 9) In the final paragraph of the results section on page 9, the authors state that 5% hexanediol alters the phase separation of Kar9, Bim1, and Bik1 in cells. But the authors only observe Kar9 in these experiments. It is possible (though maybe not probable) that Bik1 and Bim1 form their own condensate that is not affected by hexanediol. Thus, the authors should either repeat these experiments with labeled Bim1 and Bik1 or adjust the text to reflect that Kar9 no longer coalesces.
- 10) Please use magenta and green instead of red and green for the figures to enable color blind readers to interpret the presented data.

Reviewer #3:

Remarks to the Author:

Meier and co-workers describe a thorough study of three microtubule plus end-associated proteins (+TIPs), Kar9, Bim1, and Bik1, that are important for positioning the mitotic spindle properly in the bud neck in *S. cerevisiae*. Experiments with the purified proteins show that when all three are mixed together, at micromolar concentrations and in buffers with physiological levels of salt, they can form liquid-like condensates. Similar to other proteins that coacervate, their condensation in vitro is promoted by a multiplicity of interactions, including three self-associating regions identified in a prior structural analysis of Kar9. In cells sensitized for spindle positioning defects (via deletion of BIK1 or DYN1), mutating combinations of these Kar9 self-associating regions reduces cell viability, dramatically increases the probability of spindle mis-positioning, reduces the amount of Kar9 that localizes specifically to plus ends of cytoplasmic microtubules from the old spindle pole body, and

interferes with the ability of Kar9 puncta to track with the disassembling ends of these microtubules. Prior work indicates that the plus end localization of Kar9 is important for proper spindle positioning, so the reduced plus end localization in these mutants potentially explains why they cause mis-positioning. It is further shown here that the amount of Kar9 localized in the plus end puncta can vary widely, by almost 1,000-fold, depending on gene copy number and promoter strength, and that multiple puncta in a single cell can sometimes be seen to coalesce, consistent with the possibility that self-association *in vivo* creates a dynamic, non-stoichiometric condensate not strictly limited by microtubule tip area. These observations do not prove but nevertheless make a compelling case that liquid-like condensation might be important for the *in vivo* function of these +TIPs in spindle positioning. Overall the experiments are described clearly and the paper was fun to read. We support publication in Nature Cell Biology provided that the following concerns are addressed.

Primary concerns

A more substantial comparison to other liquid-liquid phase separating (LLPS) systems is needed. Do Kar9, Bim1, or Bik1 have any homologies to other proteins known to undergo LLPS? If no such homologies exist, then including at least some simple sequence analyses of their likely regions of secondary structure, probable disordered regions, and regions of low-complexity sequence would be important.

The one series of images in figure 7 panel A is insufficient to convince us that we are actually observing fusions of liquid-like condensates *in vivo*. A truly convincing case would require quantitative analysis. For instance, would the Kar9 triple mutant exhibit a different propensity for *in vivo* fusions?

The 5% hexanediol treatment (used in figure 7) is very crude and cannot provide additional convincing evidence of liquid-like phase separation *in vivo*. Hexanediol at this high concentration interferes globally with membrane permeability (Kroschwald 2017 Matters) and with kinase and phosphatase activities (Duster 2021 J Biol Chem), both of which can cause massive disruptions to intracellular organization independent of LLPS.

What are the concentrations of the proteins in the "light" phase (i.e., outside of the droplets) and do they exchange with the proteins inside the droplets? A fundamental aspect of phase separations, the exchange of components between the two coexisting phases, is not examined or even mentioned here. With some of the combinations of proteins, such as with Kar9 alone or with Kar9+Bim1, the observed condensates do not appear to be dynamic at all and instead may be merely aggregates that, given enough time, would precipitate essentially all of the protein out of solution.

Because Kar9+Bim1 seems to form static aggregates, rather than liquid-like phase separated droplets, it is confusing to include this combination in the phase diagram of figure 1 panel D. The sole example of a supposed fusion of Kar9+Bim1 condensates (shown in the second row of panel C, with one relaxation time τ plotted in panel F) is not convincing that a phase separation occurs with this particular combination. While the main text states explicitly that Kar9 alone does not form a liquid-like condensate, the behavior of Kar9+Bim1 is less clearly explained. It is confusing that Kar9+Bim1 are included in the same phase diagrams when they are exhibiting a very different behavior than the liquid-like condensates.

Minor comments and concerns

Figure 1: The ability of Bim1 (with Atto label) to disrupt droplets of Bik1 is interesting. If possible it would be interesting to see this in time-lapse. Also partial FRAP of portions of the liquid-like droplets would be interesting to see.

Figure 1: Several bands are visible near the expected molecular weight for Kar9 under the Kar9-alone and Kar9+Bik1 conditions. Is this multiplicity of bands caused by protein degradation? Or could some aggregation have occurred even under the SDS-PAGE conditions? The multiple bands for Kar9-alone and Kar9-Bik1 is unlike the triple-protein condition or the Kar9-Bim1 condition, where single clean bands are seen. Why is the behavior of Kar9 on these gels different for different combinations of proteins?

Figure 2: It would be nice here to highlight the interacting residues in these cartoon diagrams.

Figure 4: The viability assays are over-categorized, with six different color-coded categories -- far too many for a simple qualitative judgement of viability, "by eye". Moreover, the few plate images shown in the supplement do not necessarily agree with the heat map provided in the main figure. For example, the WT 35-degree plate does not look significantly worse than the corresponding gamma-alone or beta-gamma mutant plates, however the figure 4A heat map suggests a difference.

Figure 4 panel C: What cues were used to decide when a cell was in telophase for the analysis shown here? Also, please indicate the growth temp for the spindle positioning analysis somewhere in the figure or legend.

Figure 6: Why aren't any of the Kar9 mutants included here without over-expression?

TITLE – should be no more than 100 characters including spaces, without punctuation and avoiding technical terms, abbreviations, and active verbs.

Methods should be written concisely, but should contain all elements necessary to allow interpretation and replication of the results. As a guideline, Methods sections typically do not exceed 3,000 words. The Methods should be divided into subsections listing reagents and techniques. When citing previous methods, accurate references should be provided and any alterations should be noted. Information must be provided about: antibody dilutions, company names, catalogue numbers and clone numbers for monoclonal antibodies; sequences of RNAi and cDNA probes/primers or company names and catalogue numbers if reagents are commercial; cell line names, sources and information on cell line identity and authentication. Animal studies and experiments involving human subjects must be reported in detail, identifying the committees approving the protocols. For studies involving human subjects/samples, a statement must be included confirming that informed consent was obtained. Statistical analyses and information on the reproducibility of experimental results should be provided in a section titled "Statistics and Reproducibility".

All Nature Cell Biology manuscripts submitted on or after March 21 2016 must include a Data availability statement as a separate section after Methods but before references, under the heading "Data Availability". For Springer Nature policies on data availability see

<http://www.nature.com/authors/policies/availability.html>; for more information on this particular policy see <http://www.nature.com/authors/policies/data/data-availability-statements-data-citations.pdf>. The Data availability statement should include:

- Accession codes for primary datasets (generated during the study under consideration and designated as "primary accessions") and secondary datasets (published datasets reanalysed during the study under consideration, designated as "referenced accessions"). For primary accessions data should be made public to coincide with publication of the manuscript. A list of data types for which submission to community-endorsed public repositories is mandated (including sequence, structure, microarray, deep sequencing data) can be found here <http://www.nature.com/authors/policies/availability.html#data>.
- Unique identifiers (accession codes, DOIs or other unique persistent identifier) and hyperlinks for datasets deposited in an approved repository, but for which data deposition is not mandated (see here for details <http://www.nature.com/sdata/data-policies/repositories>).
- At a minimum, please include a statement confirming that all relevant data are available from the authors, and/or are included with the manuscript (e.g. as source data or supplementary information), listing which data are included (e.g. by figure panels and data types) and mentioning any restrictions on availability.
- If a dataset has a Digital Object Identifier (DOI) as its unique identifier, we strongly encourage including this in the Reference list and citing the dataset in the Methods.

We recommend that you upload the step-by-step protocols used in this manuscript to the Protocol Exchange. More details can be found at www.nature.com/protocolexchange/about.

All imaging data should be accompanied by scale bars, which should be defined in the legend. Cropped images of gels/blots are acceptable, but need to be accompanied by size markers, and to retain visible background signal within the linear range (i.e. should not be saturated). The boundaries of panels with low background have to be demarked with black lines. Splicing of panels should only be considered if unavoidable, and must be clearly marked on the figure, and noted in the legend with a statement on whether the samples were obtained and processed simultaneously. Quantitative comparisons between samples on different gels/blots are discouraged; if this is unavoidable, it should only be performed for samples derived from the same experiment with gels/blots were processed in parallel, which needs to be stated in the legend.

Figures should be provided at approximately the size that they are to be printed at (single column is 86 mm, double column is 170 mm) and should not exceed an A4 page (8.5 x 11"). Reduction to the scale that will be used on the page is not necessary, but multi-panel figures should be sized so that the whole figure can be reduced by the same amount at the smallest size at which essential details in each panel are visible. In the interest of our colour-blind readers we ask that you avoid using red and

green for contrast in figures. Replacing red with magenta and green with turquoise are two possible colour-safe alternatives. Lines with widths of less than 1 point should be avoided. Sans serif typefaces, such as Helvetica (preferred) or Arial should be used. All text that forms part of a figure should be rewritable and removable.

SUPPLEMENTARY INFORMATION – Supplementary information is material directly relevant to the conclusion of a paper, but which cannot be included in the printed version in order to keep the manuscript concise and accessible to the general reader. Supplementary information is an integral part of a Nature Cell Biology publication and should be prepared and presented with as much care as the main display item, but it must not include non-essential data or text, which may be removed at the editor's discretion. All supplementary material is fully peer-reviewed and published online as part

of the HTML version of the manuscript. Supplementary Figures and Supplementary Notes are appended at the end of the main PDF of the published manuscript.

The total number of Supplementary Figures (not including the “unprocessed scans” Supplementary Figure) should not exceed the number of main display items (figures and/or tables (see our Guide to Authors and March 2012 editorial <http://www.nature.com/ncb/authors/submit/index.html#suppinfo>; <http://www.nature.com/ncb/journal/v14/n3/index.html#ed>). No restrictions apply to Supplementary Tables or Videos, but we advise authors to be selective in including supplemental data.

GUIDELINES FOR EXPERIMENTAL AND STATISTICAL REPORTING

REPORTING REQUIREMENTS – We are trying to improve the quality of methods and statistics reporting in our papers. To that end, we are now asking authors to complete a reporting summary that collects information on experimental design and reagents. The Reporting Summary can be found here <https://www.nature.com/documents/nr-reporting-summary.pdf> If you would like to reference the guidance text as you complete the template, please access these flattened versions at <http://www.nature.com/authors/policies/availability.html>.

Information on how many times each experiment was repeated independently with similar results needs to be provided in the legends and/or Methods for all experiments, and in particular wherever

representative experiments are shown.

Author Rebuttal to Initial comments

I should stress that the referees' concerns point to unclear assessment of Kar9 subcellular localization and behaviour, as well as unclear assessment of biomolecular condensation which would need to be addressed with experiments and data, and reconsideration of the study for this journal and re-engagement of referees would depend on strength of these revisions.

In particular, it would be essential to:

A) Assess Kar9 subcellular localization (Reviewers #1 and #2)

As requested by the reviewers, we have now added quantifications for the distribution of Kar9 between the two asters (Figure 5B) and of the total amount of microtubule-bound Kar9 (Supp. Figure 7) for the key strains, namely WT, *kar9- $\alpha\beta\gamma^-$* , *bik1 Δ* singles and *kar9- $\alpha\beta\gamma^-$ bik1 Δ* double mutant. We now also provide the frequency of +TIP splitting and fusion in living cells (Figure 7B), as a function of genotype in the same mutant cells as above. Together, these data provide a much sharper representation of the subcellular localization of Kar9 and of the behavior of the +TIP body

B) Completely characterize the biomolecular condensate behaviour of, and effects of, Kar9 and Bim1 and Bik1 (all Reviewers)

We have added FRAP experiments to document the dynamics of Kar9, Bim1, and Bik1 in the different condensates (Figure S1) as well as quantification of their partitioning coefficients (Figure S2A). We have also attempted to characterize the phase diagram of the different mutant alleles of Kar9 together with Bik1 alone, as requested by reviewer 2.

However, after many different attempts we were not able to produce these mutant proteins in stable amounts necessary for doing these experiments. These mutant proteins fall out of solution too rapidly for us to produce phase diagrams with them. In vivo, Kar9 and Bim1 interact with each other with low nanomolar affinity (Manatschal et al., 2016), such that Kar9 is always bound to Bim1 in the cell. This is consistent with the fact that Kar9 does not localize to microtubules at all in the absence of Bim1 (see for example Korinek et al., 2000). Thus, Kar9 and Bim1 can be legitimately considered as an obligate, single unit in vivo. In this context, making the phase diagram of the different alleles of Kar9 with Bik1 alone makes sense neither from an experimental nor from biological and functional points of view. We now explain this in the rebuttal and the main text.

C) Investigate the regulatory relationship between Kar9, Bik1, and Bim1 behaviour on microtubule spindles (all Reviewers)

In Figure 5F-G and S8 we now provide a more complete analysis of microtubules and Kar9 puncta dynamics and the effects of the different mutations, namely the *kar9- $\alpha\beta\gamma$ ⁻* and *bik1 Δ* alleles and their combination. These data provide a much sharper representation of the dynamics of Kar9 and the +TIP-body in vivo, and of the effects of affecting self-interaction interfaces in the +TIP-body. Together, these studies strengthen the notion that self-interaction among all two main components, namely the Kar9/Bim1 unit and Bik1, is essential for the assembly, liquid behavior, and function of the Kar9 +TIP body.

In Figure 7C-E and Figure S9, we have expanded our analysis of the effect of hexanediol on the Kar9 +TIP body by establishing that it has the same effect on Kar9, Bim1, and Bik1, but does not affect microtubule organization in the short time and conditions that we used. Thus, these data confirm that the entire body disassembles at once and specifically.

D) All other referee concerns pertaining to strengthening existing data, providing controls, methodological details, clarifications and textual changes, should also be addressed.

We went carefully through all requests of the reviewers and addressed them all, in full. We have also extensively edited the figures and the text to increase clarity.

E) Finally please pay close attention to our guidelines on statistical and methodological reporting (listed below) as failure to do so may delay the reconsideration of the revised manuscript. In particular please provide:

We would be happy to consider a revised manuscript that would satisfactorily address these points, unless a similar paper is published elsewhere, or is accepted for publication in Nature Cell Biology in the meantime.

We would like to receive a revised submission within six months.

We hope that you will find our referees' comments, and editorial guidance helpful. Please do not hesitate to contact me if there is anything you would like to discuss.

Best wishes,

Daryl Jason David

In summary, the recommendations of the reviewers have helped to greatly improve our manuscript, which now much more clearly than before demonstrate the assembly of the +TIP network into a liquid-like droplet at the plus-end of microtubules, and the importance of this condensation process for the function of this +TIP body. We hope that you will now find our manuscript suitable for publication.

With our best regards,

Yves and Michel

Reviewers' Comments:

Reviewer #1:

Remarks to the Author:

In their manuscript, Meier et al. discover a network of proteins that form what they coin the “+TIP body”, and show that the underlying mechanisms that lead to the formation of this body is liquid-liquid phase separation. Microtubules are evolutionary conserved cytoskeletal found in all eukaryotes. They interact with a multitude of different proteins of which more and more were discovered over the past decades. Given this multitude of interactors, plus the fact that microtubules are highly similar within a single cell, one of the burning questions in cell biology is how single microtubules can acquire specific functions that clearly distinguish them from their neighbours. One intuitive mechanism is that they are able to acquire a unique set of interacting proteins, which in turn assure their functional specialization. The current manuscript demonstrates how low-affinity interactions between multiple proteins at the plus-tip (+TIP) of microtubules forms by liquid-liquid phase separation, and how this “Kar9-body” determines cytoplasmic microtubule functions.

Liquid-liquid phase separation has been intensely studied in the past years, however, direct functional studies in cells have remained rather scarce. The current work takes an important step toward determining the biological role of liquid-liquid phase separation in living cells. Using yeast as a model system has a number of important advantages, out of which the most important might be the small number of microtubules in yeast cells. This provides the great advantage that the function of single, well-defined microtubules can be characterized, which would be immensely more difficult in larger cells, such as mammalian cells, which have thousands of microtubules. The results of the current study, by contrast, are likely to apply to many different cells and organisms, as the proteins that form the Kar9-body are evolutionarily conserved.

The authors first characterise the phase-separation behaviour of three components of the +TIP network, Bim1, Bik1 and Kar9 using purified proteins. They demonstrate that phase separation of Bim1 and Bik1 is promoted by Kar9, which alone does not phase-separate but precipitates. Having defined the precise biochemical conditions required for the formation of a phase-separated “Kar9-body”, the authors perform a series of site-specific mutagenesis studies to define the interaction sites in the proteins necessary for their interactions, and thus, for the formation of condensates.

Next the authors use the power of yeast genetics to define the importance of their biochemically identified mechanisms in vivo. Mutating the amino acid residues that had been biochemically defined as interaction surfaces, the authors test which of these mutations affect the viability of the yeast cells. They find that only mutations of three interaction surfaces of Kar9 have a strong impact on cell viability, and only when Bik1 is deleted (Bik1Δ). The most striking phenotype is spindle misalignment.

Next, they demonstrate that the Kar9-Bik1 interface is important for the microtubule-specific localisation of the “Kar9-body” in the cell, and that perturbation of the interfaces lead to a partial loss of the characteristic back-tracking of this body on shrinking microtubules. This phenotype is further amplified when Bik1 is lacking. Finally, the authors demonstrate that the size of the Kar9-body changes with the amount of available protein in the cell, which indicates that it is generated by phase separation as is not the case for defined protein complexes. This is elegantly shown by comparison with the kinetochore protein Dam1. The authors further show examples of fusing Kap9 drops on microtubules, and the dissolution of the drops in hexanediol, both indications for their liquid character.

The paper is clearly written, the experiments are meticulously performed and analysed, and figures are in most cases clear (apart from some points listed below). The observations reported in this manuscript are important and groundbreaking as they provide a novel model for the functioning of +TIP tracking based on liquid-liquid phase separation. +TIP tracking of a number of proteins on growing microtubules plays key roles in a number of microtubule-dependent biological processes, and one of the unanswered questions was how a highly divergent group of proteins (the +TIPs) could achieve the same biophysical behaviour. The discoveries in the current paper suggest that liquid-liquid phase separation, based on multivalent but not necessarily specific interactions could be a common theme in this process. The novelty and broad importance of these findings makes this manuscript a strong candidate for Nature Cell Biology.

We thank the reviewer for these very positive and enthusiastic comments and for the very useful suggestions that we have addressed in full.

Major Points:

1) In Fig. 5A, the authors quantify the intensity of the Kar9 fluorescence intensity – supposedly at the +TIPs of astral microtubules where it is normally localised? This is not clear and should be better specified in the text.

Yes, as we showed before, Kar9 localizes to the tip of astral microtubules or to the SPB when these microtubules shrink. We have now stated more precisely the localization of Kar9 in the revised manuscript (see page 8, fourth paragraph).

In Fig. 5B they then show that the mutations lead to a re-localisation of Kar9 to the other side of the spindle where it is not supposed to localise. To show this, however, the authors use a more qualitative analyses than in panel A. Given that they are able to quantify Kar9-mNeonGreen intensities, why don't they connect the data in panel A and B to a single analysis where they show the re-distribution of Kar9 from one to two spindle poles in a quantitative manner.

The readout could for instance be the fluorescence intensity ration between the two poles, instead of a rather arbitrary weakly / strongly asymmetric term currently used in panel B. A second parameter analysed could be the overall fluorescence of Kar9 on BOTH poles (as both represent +TIP localisation), and not only one pole as currently in panel A.

We have followed this recommendation and are now reporting the distribution of Kar9 in a more quantitative manner in Figure 5B. We have opted for an “asymmetry index”, which reports the difference between the most and least intensely labelled spindle poles divided by the sum of both. This index distributes between 0 and 1. This is the type of quantification that we had been using in the past (see, for example, (Chen et al., 2019; Hotz et al., 2012; Lengefeld et al., 2017)). Zero indicates that both asters recruit the same amount of Kar9 and 1 indicates that Kar9 is all localized to one single spindle pole. Note that in some instances, background subtraction has led to small negative intensities on poles lacking Kar9 and hence to a few values slightly above 1.

We are now also providing the information about the distribution for the sum intensity for some of the key mutant combinations in supplementary Figure S7.

Minor points:

1) In the introduction, the authors write: “...spindle pole bodies (SPBs, equivalent of the metazoan centrosomes)”. The SPBs are functional equivalents of centrosomes, however not structurally. This should be pointed out for the readers that are outside the centrosome and cytoskeleton field.

We are now pointing this out more carefully in the Introduction section of our revised manuscript (page 3, third paragraph).

2) Fig. 1A, 2D, 7E: It is well-appreciated that the authors use schematic drawings to guide the reader. However, they might want to use the same colour code for all the figures in the paper (compare Fig. 1A and 2D with Fig. 7E). They might also want to show their proteins approximately to scale: Bim1 is about half the size of Bik1 and Kar9, but on their drawings they are all the same size.

Thank you for these very useful suggestions. We have now unified our color code throughout the manuscript and adapted the size of the molecules in the drawing to better reflect their relative sizes.

3) In several figures, the authors use a simplified nomenclature for their mutations. Though this is somewhat

explained in the figure legends, the authors might consider having a small legend directly within the figure where this nomenclature is explained, this makes the reading of the figure easier.

We have now adapted our nomenclature to make it clearer, more intuitive, and better explained in both the revised text and the figures.

4) Fig. 4: It is very difficult to read this figure; the authors should make an effort to guide the reader. For instance: how do the data in panel A relate to the plots shown in panels C & D: is for instance what is labelled as tree spots in panel A the same as Kar9 followed by 3 spots in the plots?

The new nomenclature addresses this point. We have also been more careful with making sure that the same labelling is used throughout the figures.

*Moreover, the authors could use bars below the legend to show which subset of the diagram represents *bik1Δ* and *dyn1Δ* in order to make it more comparable to A.*

Thank you, this is indeed a very useful suggestion and facilitates the reading of the figure a lot. We have also adapted all figures accordingly.

In the microscopy images, increasing contrast of the red channel plus a schematic drawing representing the phenotypes would improve the understandability of the figure.

We have added a drawing below the figures and improved the contrast of the images.

5) The authors should decide if they want to call their discovery “Kar9-body” (used throughout the text) or “+TIP-body” as used in the title.

Thank you for raising this point. We settled on +TIP-body and edited the text accordingly.

Reviewer #2:

Remarks to the Author:

In the article entitled “High interaction valency ensures cohesion and persistence of a microtubule +TIP body at the plus-end of a single specialized microtubule in yeast,” by Meier, Farcas, and colleagues, the authors investigate the effect of protein valency on microtubule plus-end condensate formation and persistence. The authors focus specifically on Kar9, Bim1, and Bik1. In vitro reconstitution assays using Kar9, Bim1, and Bik1 demonstrate that each component contributes to condensate formation and phase behavior. The authors used X-ray crystallography to identify three conformations that enabled Kar9 dimerization, each with a unique binding interface. Mutation of individual interfaces altered the ability of Kar9 to undergo phase separation while mutations of combinations of interfaces had greater effect. Mutations in Kar9 also affect its ability to undergo phase separation with Bik1 and Bim1. In yeast in which Bik1 was deleted, combinations of Kar9 mutations resulted in misalignment of the mitotic spindle and increased cell death at increasing temperatures. Mutations in Kar9 also reduced the amount of Kar9 at microtubule plus-ends, and mutation of all three interfaces decreased Bik1 levels at microtubule plus-ends. Finally, the authors connect the valency of plus-end binding partners to microtubule dynamics and the liquid-like nature of plus-end condensates.

Overall, this reviewer found this manuscript to be well organized and the presented data to be excellent. The authors observations and investigation of specific binding surfaces of Kar9 and their analysis of Kar9 interactions provide important insights into the multivalent network of proteins at the plus-ends of microtubules. Their cellular experiments do a fine job at connecting in vitro observations with cellular function.

We are very thankful to the reviewer for these very positive comments and for the suggestions that we have examined carefully.

However, there are some experimental details in the paper that are assumed by the authors but not shown in the data. For example, phase separation assays with Kar9 mutants are performed with either Bim1 and Bik1 or Bim1 alone, not Bik1 alone.

Concerning the phase separation assays of the Kar9 mutants with Bik1 alone, we agree that such data would be great to have and would make the study more complete. However, this proved impossible, for technical reasons, preventing us to generate and report reliable results. In fact, the different Kar9 variants turned out to be particularly insoluble in the absence of Bim1, preventing us from preparing the corresponding proteins in a well-behaved state, in decent amounts, and in the concentrations required for the experiments. After several months of trying and failing to solve that problem in different manners, we have decided to take and report this as a fact. This observation makes actually sense from the biochemical and functional points of view. Kar9 interacts with nanomolar affinity with Bim1 in vitro (Manatschal et al., 2016), suggesting that it is never dissociated from it in living cells. Therefore, it makes

sense to consider these two proteins as an obligate, single unit. We are now explaining these limitations in the revised manuscript on page 7, paragraph 4.

Additionally, some of the text should be updated using common condensate field language.

We have edited the text accordingly, to the best of our knowledge, and along the examples proposed by the reviewer below.

These comments will be addressed in greater detail below. If the authors can address the concerns outlined by this reviewer, this paper will be a good candidate for publication in Nature Cell Biology.

We hope that the reviewer will understand the limitations elaborated above and find the new version of the manuscript now suitable for publication.

Major Comments:

1) Based on the authors data, Kar9 can self-associate into three different dimer conformations. Do the authors have any insights into the relative ratio of these interactions within their crystal lattices? Is one conformation preferred over the others?

As reported previously, our Kar9 crystal belongs to space group $P4_1$ and contains two Kar9 protomers per asymmetric unit (Kumar et al., 2021). All contract points in the crystal are equally represented due to symmetry considerations. Thus, no dimer conformation is preferred over the others.

Relatedly, does one of the conformations promote network / condensate formation?

Our data establish that all three interfaces are redundant with each other but α is the first contact to form, consistent with the fact that it is the highest affinity interface (Kumar et al., 2021). Accordingly, variants carrying the alpha mutations generally cause stronger phenotypes than the β and γ mutations. We thus assume that the α interface initiates the phase separation process (see bottom of page 6 in the revised manuscript).

Also, are these conformations kinetically trapped, or can they exchange between different conformations? Perhaps using their existing mutation data, the authors can provide insight into these questions.

We have now added FRAP data to address these questions. The data show that the proteins indeed exchange, but interestingly not all at the same speed. Bik1 exchanges faster than Kar9 and Bim1, with the latter two showing similar dynamics, consistent with Kar9 and Bim1 being tightly bound to each other (as mentioned above and discussed; see middle paragraph of page 7). The observed exchange of Kar9 suggests that no stable, kinetically trapped lattice exists, like in the crystal, but that there is exchange and competition between conformations in the context of full-length proteins inside condensates.

2) As stated above, the data in Figure 3 isn't complete. The authors state in the concluding paragraph of the section entitled "Role of Kar9 self-association for phase separation of the core Kar9-network components in vitro" that Bik1 and Kar9 interactions sites contribute to phase separation and that Bik1, Bim1, and Kar9 interactions are redundant in driving phase separation. However, the authors never test Kar9 and Bik1 alone. They should run a set of experiments to examine Kar9 behavior with Bik1 in Figure 3 to provide a complete picture of the behavior of this interaction network.

As we mentioned in the response to the general comments of the reviewer above, and despite many attempts to overcome the issue, we were not able to generate well-behaved, stable, and soluble forms of the Kar9 variants in the absence of Bim1. It should be noted that even for wild type Kar9, we observed that Kar9 is not as stable in the absence of Bim1 as in its presence. This is illustrated, e.g., by the presence of degradation products in the gel shown in Figure 1A for Kar9 purified alone. Given the nanomolar affinity of Kar9 and Bim1 for each other (Manatschal *et al.*, 2016), Kar9 might always be associated with Bim1 in vivo (there is large excess of Bim1 over Kar9 in the cell, with an 8:1 ratio; SGD data base). This might explain why wild type Kar9 is less stable when purified in the absence of Bim1 and the situation is even worse for the Kar9 interface mutants. In this perspective, the Kar9-Bim1 complex should be considered as part of an obligate, single unit, a view that is supported by the fact that the *kar9Δ bim1Δ* double mutant is not sicker than the *bim1Δ* and *kar9Δ* single mutants (Korinek *et al.*, 2000).

Minor Comments:

1) Throughout the manuscript, condensate language / jargon should be used. For example, instead of "solubilized in these droplets" use enriched in these droplets; instead of dots use puncta, etc.

We have edited the text accordingly.

2) *Figure 1A can be removed from the manuscript. It doesn't add to an understanding of the proteins tested and is mostly redundant (and less informative) with Figure 2D.*

This is a good point and we followed your advice.

3) *In the first paragraph of the first results section, the authors state that Kar9 collapses into amorphous aggregates. The description should be changed because the biophysical properties of these droplets is not tested. These structures are consistent with rapidly maturing, gel-like condensates that have been frequently observed (See PMID 33148640).*

We have now replaced the word aggregate with precipitates, which is indeed more neutral and more accurate.

4) *Why must Bik1 contain three homodimerization domains (middle of page 6)?*

We removed this statement and instead now state that Bik1 contains multiple interaction domains.

5) *Figure 2E (middle of page 6) should be Figure 2D.*

Thank you for catching this. We have now verified again that all figures are properly cited in the revised manuscript.

6) *In Figure 2D, please include details of domains of Kar9, Bim1, and Bik1 in the figure.*

As suggested, we now provide more detailed information on the domain organization of the proteins in the revised Figure 2A. This indeed makes the paper more understandable for a broad audience.

7) *The layout of Figure 3A is a bit confusing. It would be helpful if both WT experiments were represented by a dotted line and both mutant experiments were filled in as they currently are. By having 1 WT experiment filled and one dotted, it took a bit of consideration to understand what the authors were showing the reader.*

Thank you for this comment. We have now followed the recommendation of the reviewer and improved our representation to increase clarity. We now also provide a schematic explanation of the phase diagrams in Figure 1C to help readers who are not familiar with them.

8) *At the top of page 8, the authors comment that alpha and beta mutants reduce Kar9 levels at microtubule plus-ends by 40% and 20%, respectively. In reviewing their data, it seems that these reductions have minimal impact on cellular function, which strongly suggests that it is the formation of a Kar9 condensate that is important rather than the amount of Kar9 at the plus-end. This is consistent with other observations that correlate phase separation of molecules to cellular function rather than the discrete number of molecules (See PMID 27056844). It would be good for the authors to emphasize this point because it is quite possibly general principle of different phase separating systems.*

This is indeed an excellent point that we now included in the Discussion section of the revised manuscript (page 11, middle).

9) *In the final paragraph of the results section on page 9, the authors state that 5% hexanediol alters the phase separation of Kar9, Bim1, and Bik1 in cells. But the authors only observe Kar9 in these experiments. It is possible (though maybe not probable) that Bik1 and Bim1 form their own condensate that is not affected by hexanediol. Thus, the authors should either repeat these experiments with labeled Bim1 and Bik1 or adjust the text to reflect that Kar9 no longer coalesces.*

Thank you for this recommendation. We have now added these data in the new version of Figure 7E. What we found is that Kar9, Bim1, and Bik1 disappear together from microtubule tips. We have also added the analysis of the effect of hexanediol on astral microtubules, which is very minor (see Figure S10B-C). Thus, at the early time point that we took, the effect seems to affect the entire +TIP body specifically and not the underlying microtubules as much.

10) *Please use magenta and green instead of red and green for the figures to enable color blind readers to interpret the presented data.*

We now have consistently used magenta and green for all fluorescence microscopy data.

Reviewer #3:

Remarks to the Author:

Meier and co-workers describe a thorough study of three microtubule plus end-associated proteins (+TIPs), Kar9, Bim1, and Bik1, that are important for positioning the mitotic spindle properly in the bud neck in S. cerevisiae. Experiments with the purified proteins show that when all three are mixed together, at micromolar concentrations and in buffers with physiological levels of salt, they can form liquid-like condensates. Similar to other proteins that coacervate, their condensation in vitro is promoted by a multiplicity of interactions, including three self-associating regions identified in a prior structural analysis of Kar9. In cells sensitized for spindle positioning defects (via deletion of BIK1 or DYN1), mutating combinations of these Kar9 self-associating regions reduces cell viability, dramatically increases the probability of spindle mis-positioning, reduces the amount of Kar9 that localizes specifically to plus ends of cytoplasmic microtubules from the old spindle pole body, and interferes with the ability of Kar9 puncta to track with the disassembling ends of these microtubules. Prior work indicates that the plus end localization of Kar9 is important for proper spindle positioning, so the reduced plus end localization in these mutants potentially explains why they cause mis-positioning. It is further shown here that the amount of Kar9 localized in the plus end puncta can vary widely, by almost 1,000-fold, depending on gene copy number and promoter strength, and that multiple puncta in a single cell can sometimes be seen to coalesce, consistent with the possibility that self-association in vivo creates a dynamic, non-stoichiometric condensate not strictly limited by microtubule tip area. These observations do not prove but nevertheless make a compelling case that liquid-like condensation might be important for the in vivo function of these +TIPs in spindle positioning. Overall the experiments are described clearly and the paper was fun to read. We support publication in Nature Cell Biology provided that the following concerns are addressed.

We thank the reviewer for these very supportive comments and the very useful suggestions listed below. We are very happy that the reviewer is so positive about our observations.

Primary concerns

A more substantial comparison to other liquid-liquid phase separating (LLPS) systems is needed. Do Kar9, Bim1, or Bik1 have any homologies to other proteins known to undergo LLPS? If no such homologies exist, then including at least some simple sequence analyses of their likely regions of secondary structure, probable disordered regions, and regions of low-complexity sequence would be important.

Thank you for this suggestion. The domain organizations of Bim1, Bik1, and Kar9 are now described in detail on pages 5 (bottom) and 6 (top). We also mention that their homologs have been found to undergo phase separation in co-submitted papers (Maan et al., 2021) (Song et al., 2022) and a paper in revision elsewhere (Miesch et al., 2022). A supplementary figure outlining the disordered domain predictions for all three proteins is now also provided (new Figure S3). Note that the proposed homologs of Kar9 (MACF, SLAIN, and APC; (Bienz, 2001; Kumar *et al.*, 2021; Manatschal *et al.*, 2016)) are not known to phase separate at this stage.

The one series of images in figure 7 panel A is insufficient to convince us that we are actually observing fusions of liquid-like condensates in vivo. A truly convincing case would require quantitative analysis. For instance, would the Kar9 triple mutant exhibit a different propensity for in vivo fusions?

This is a good point. We have now provided measurements of the frequency of puncta fusion and fission (new Figure 7B). These data show that fusion is not a rare event (one event every 2.5 minutes on average in wild type cells). In wild type cells, the frequency of fusion is twice as high as that of fission, as would be expected, given that fission of a droplet is energetically expensive in contrast to the spontaneous reverse reaction of fusion. In the triple mutant cells, both frequencies increase and the relative frequencies of fusion and fission become more similar. These effects are exacerbated upon further removal of the *BIK1* gene. These data are consistent with the cohesion of the droplets being reduced in these mutant cells. We describe these data in the revised manuscript (top of page 9).

The 5% hexanediol treatment (used in figure 7) is very crude and cannot provide additional convincing evidence of liquid-like phase separation in vivo. Hexanediol at this high concentration interferes globally with membrane permeability (Kroschwald 2017 Matters) and with kinase and phosphatase activities (Duster 2021 J Biol Chem), both of which can cause massive disruptions to intracellular organization independent of LLPS.

We agree that in their original form, these data were not as convincing for the reasons mentioned by the reviewer. However, the tight correlation between the effect of hexanediol on the +TIP body in vitro (Figure 7C) and in vivo (Figure 7D-E and S10) and the fact that the effects that we report were observed within a few minutes of the treatment lead us to characterize this experiment further. We now show that hexanediol has little effect on the presence of astral microtubules at the time point of observation, unlike nocodazole, which itself does not cause the disassembly of the +TIP-body. On the other hand, we now show that hexanediol also causes the dissociation of Bim1 and Bik1 from the plus end of the astral microtubules. We would like to emphasize that we do not know of kinases that are activated within minutes of cell stress and we have not observed that Kar9 would be displaced from microtubule tips either upon heat shock or in response to other classical stressors. Thus, our initial experiments along with the

new controls are very suggestive of the +TIP body being held together in vivo, like in vitro, by labile, low affinity interactions that are easily perturbed by hexanediol.

Nevertheless, we have rephrased our conclusions of these experiments by saying that they support the notion that formation of the +TIP body “relied on low affinity interactions driving phase-separation” (see page 10, second last paragraph).

What are the concentrations of the proteins in the "light" phase (i.e., outside of the droplets) and do they exchange with the proteins inside the droplets? A fundamental aspect of phase separations, the exchange of components between the two coexisting phases, is not examined or even mentioned here. With some of the combinations of proteins, such as with Kar9 alone or with Kar9+Bim1, the observed condensates do not appear to be dynamic at all and instead may be merely aggregates that, given enough time, would precipitate essentially all of the protein out of solution.

The gels in Figure 1A indicate that the concentration of the proteins in the light phase is low but not null. The notion that the proteins exchange with the cytoplasm is now supported by FRAP experiments. These experiments establish that Kar9 and Bim1 exchange, albeit slowly, with the light phase and move, even if slowly again, within the droplet. In contrast, we found that Bik1 exchanges substantially faster (new Figure S1).

Together, these results are consistent with the partition coefficients for Kar9, Bim1, and Bik1 being not infinitely high but in the range of 8 to 30 (new Figure S2A).

Because Kar9+Bim1 seems to form static aggregates, rather than liquid-like phase separated droplets, it is confusing to include this combination in the phase diagram of figure 1 panel D. The sole example of a supposed fusion of Kar9+Bim1 condensates (shown in the second row of panel C, with one relaxation time tau plotted in panel F) is not convincing that a phase separation occurs with this particular combination. While the main text states explicitly that Kar9 alone does not form a liquid-like condensate, the behavior of Kar9+Bim1 is less clearly explained. It is confusing that Kar9+Bim1 are included in the same phase diagrams when they are exhibiting a very different behavior than the liquid-like condensates.

FRAP experiments (Figure S1) and the movies of droplet fusion (Figure 1B and Movie S2) show beyond doubt that Kar9+Bim1 do not form static aggregates. We now chose a more convincing example of coalescing droplets. These droplets are certainly more viscous and may age more rapidly than those containing Bik1 but are not aggregates or precipitates. Due to the slow dynamics, quantifiable 1:1 droplet fusions are rare. We now have clarified this issue on page 5, paragraph 3 of the revised manuscript.

Minor comments and concerns

Figure 1: The ability of Bim1 (with Atto label) to disrupt droplets of Bik1 is interesting. If possible it would be interesting to see this in time-lapse. Also partial FRAP of portions of the liquid-like droplets would be interesting to see.

Although we also find this observation interesting, a more detailed analysis of the effect of Bim1 on Bik1 droplets is not within the scope of this paper. Nevertheless, we have now added representative movies of the dissolution of Bik1 droplets by Bim1 as supplementary data for the curiosity of the reader (supplementary movie S5).

Figure 1: Several bands are visible near the expected molecular weight for Kar9 under the Kar9-alone and Kar9+Bik1 conditions. Is this multiplicity of bands caused by protein degradation? Or could some aggregation have occurred even under the SDS-PAGE conditions? The multiple bands for Kar9-alone and Kar9-Bik1 is unlike the triple-protein condition or the Kar9-Bim1 condition, where single clean bands are seen. Why is the behavior of Kar9 on these gels different for different combinations of proteins?

As mentioned above, Kar9 is less stable in absence of Bim1. For most experiments, we solved this problem by co-purifying it together with Bim1. For experiments that require the absence of Bim1, we had to use separately purified Kar9, which has more degradation problems as can be seen in Figure 1A. The multiplicity of bands is not caused by the pelleting experiment. In our hands, the differences in protein quality do not significantly change the phase behavior, most likely because truncated proteins contribute to phase separation like the full length protein does.

Figure 2: It would be nice here to highlight the interacting residues in these cartoon diagrams.

We have now highlighted the interacting residues that we mutated to target each interface as magenta spheres in Figure 2C.

Figure 4: The viability assays are over-categorized, with six different color-coded categories - far too many for a simple qualitative judgement of viability, "by eye". Moreover, the few plate images shown in the supplement do not necessarily agree with the heat map provided in the main figure. For example, the WT 35-degree plate does not look significantly worse than the corresponding gamma-alone or beta-gamma mutant plates, however the figure 4A heat map suggests a difference.

This is a helpful point. We have done so now, and reduced our code to four color categories. This simplifies the data a lot. We apologize for the discrepancy between the heat map and the images. We do observe some mild variability from isolate to isolate, explaining the differences. Reduction to four categories solves that problem.

Figure 4 panel C: What cues were used to decide when a cell was in telophase for the analysis shown here? Also, please indicate the growth temp for the spindle positioning analysis somewhere in the figure or legend.

The word telophase was replaced with “late anaphase cells” and we explain that this refers to cells where the spindle length reaches a least the diameter of the mother cell (see Methods section, “microscopy data analysis”, first paragraph).

Figure 6: Why aren't any of the Kar9 mutants included here without over-expression?

The corresponding data are already presented in Figure 5A.

References

- Bienz, M. (2001). Spindles cotton on to junctions, APC and EB1. *Nat Cell Biol* 3, E67-68. [10.1038/35060140](https://doi.org/10.1038/35060140).
- Chen, X., Widmer, L.A., Stangier, M.M., Steinmetz, M.O., Stelling, J., and Barral, Y. (2019). Remote control of microtubule plus-end dynamics and function from the minus-end. *Elife* 8. [10.7554/eLife.48627](https://doi.org/10.7554/eLife.48627).
- Hotz, M., Leisner, C., Chen, D., Manatschal, C., Wegleiter, T., Ouellet, J., Lindstrom, D., Gottschling, D.E., Vogel, J., and Barral, Y. (2012). Spindle pole bodies exploit the mitotic exit network in metaphase to drive their age-dependent segregation. *Cell* 148, 958-972.
- Korinek, W.S., Copeland, M.J., Chaudhuri, A., and Chant, J. (2000). Molecular linkage underlying microtubule orientation toward cortical sites in yeast. *Science* 287, 2257-2259.

Kumar, A., Meier, S.M., Farcas, A.M., Manatschal, C., Barral, Y., and Steinmetz, M.O. (2021). Structure and regulation of the microtubule plus-end tracking protein Kar9. *Structure*. 10.1016/j.str.2021.06.012.

Lengefeld, J., Hotz, M., Rollins, M., Baetz, K., and Barral, Y. (2017). Budding yeast Wee1 distinguishes spindle pole bodies to guide their pattern of age-dependent segregation. *Nat Cell Biol* 19, 941-951. 10.1038/ncb3576.

Maan, R., Reese, L., Volkov, V.A., King, M.R., van der Sluis, E., Andrea, N., Evers, W., Jakobi, A.J., and Dogterom, M. (2021). Multivalent interactions facilitate motor-dependent protein accumulation at growing microtubule plus ends. *bioRxiv*, 2021.2009.2014.460284. 10.1101/2021.09.14.460284.

Manatschal, C., Farcas, A.M., Degen, M.S., Bayer, M., Kumar, A., Landgraf, C., Volkmer, R., Barral, Y., and Steinmetz, M.O. (2016). Molecular basis of Kar9-Bim1 complex function during mating and spindle positioning. *Mol Biol Cell* 27, 3729-3745. 10.1091/mbc.E16-07-0552.

Miesch, J., Wimbish, R.T., Velluz, M.-C., and Aumeier, C. (2022). Phase separation of +TIP-networks regulates microtubule dynamics. *bioRxiv*, 2021.2009.2013.459419. 10.1101/2021.09.13.459419.

Decision Letter, first revision:

12th July 2022

Dear Dr. Barral,

I apologize for the delay. Thank you for submitting your revised manuscript "Multivalency ensures cohesion and persistence of a microtubule +TIP-body at the plus-end of a single specialized microtubule in yeast" (NCB-M46766A). It has now been seen by the original referees and their comments are below. The reviewers find that the paper has improved in revision, and therefore we'll be happy in principle to publish it in *Nature Cell Biology*, pending minor revisions to satisfy the referees' final requests and to comply with our editorial and formatting guidelines.

Thank you again for your interest in *Nature Cell Biology* Please do not hesitate to contact me if you have any questions.

Sincerely,
Daryl

Daryl J.V. David, PhD

Senior Editor, Nature Cell Biology
Consulting Editor, Nature Communications
Nature Portfolio

Heidelberger Platz 3, 14197 Berlin, Germany
Email: daryl.david@nature.com
ORCID: <https://orcid.org/0000-0002-9253-4805>

Reviewer #1 (Remarks to the Author):

The authors have carefully addressed all points raised in the initial review. I have no further comments on the current manuscript.

Reviewer #2 (Remarks to the Author):

The authors have done a good job of addressing the concerns of this reviewer. The addition of an explanation for the technical difficulties encountered while attempting to address this reviewer's concerns and a reasonable explanation for these difficulties will be helpful for other groups (and even future researchers in corresponding authors' groups). The other edits and additional data added to this manuscript greatly improve the presented study. This reviewer supports publication of this manuscript in Nature Cell Biology.

Reviewer #3 (Remarks to the Author):

Meier et al have done a nice job revising their manuscript. The new version was a pleasure to read, and it provides a thorough study of both the *in vitro* and *in vivo* behaviors of the network of three important +TIP proteins, Kar9, Bim1, and Bik1. In particular, I note that the revised manuscript includes a more comprehensive discussion of the domain organization of the proteins and how these proteins compare to other proteins known to undergo liquid-liquid phase separation. I also appreciate the addition of new quantitative data establishing that fusion of Kar9 puncta *in vivo* is not rare, and that the rates of fusion and fission are accelerated in strains with mutations in the key protein-protein interaction sites within the network. My earlier concerns about hexandiol are at least partially alleviated by the data showing that about 70% of hexandiol-treated cells retain discernable astral MTs. These observations show that the treatment did not massively disrupt the microtubule cytoskeleton. New data show that levels of all three +TIPs at the ends of astral microtubules *in vivo* are strongly reduced by hexanediol and more cautious language is used now in the corresponding paragraph (lines 341-343). Overall, the manuscript is excellent and, in my view, deserves publication.

Decision Letter, final checks:

Our ref: NCB-M46766A

5th August 2022

Dear Dr. Barral,

Thank you for your patience as we've prepared the guidelines for final submission of your Nature Cell Biology manuscript, "Multivalency ensures cohesion and persistence of a microtubule +TIP-body at the plus-end of a single specialized microtubule in yeast" (NCB-M46766A). Please carefully follow the step-by-step instructions provided in the attached file, and add a response in each row of the table to indicate the changes that you have made. Please also check and comment on any additional marked-up edits we have proposed within the text. Ensuring that each point is addressed will help to ensure that your revised manuscript can be swiftly handed over to our production team.

We would like to start working on your revised paper, with all of the requested files and forms, as soon as possible (preferably within one week). Please get in contact with us if you anticipate delays.

In recognition of the time and expertise our reviewers provide to Nature Cell Biology's editorial process, we would like to formally acknowledge their contribution to the external peer review of your manuscript entitled "Multivalency ensures cohesion and persistence of a microtubule +TIP-body at the plus-end of a single specialized microtubule in yeast". For those reviewers who give their assent, we will be publishing their names alongside the published article.

Nature Cell Biology offers a Transparent Peer Review option for new original research manuscripts submitted after December 1st, 2019. As part of this initiative, we encourage our authors to support increased transparency into the peer review process by agreeing to have the reviewer comments, author rebuttal letters, and editorial decision letters published as a Supplementary item. When you submit your final files please clearly state in your cover letter whether or not you would like to participate in this initiative. Please note that failure to state your preference will result in delays in accepting your manuscript for publication.

Cover suggestions

As you prepare your final files we encourage you to consider whether you have any images or illustrations that may be appropriate for use on the cover of Nature Cell Biology.

Nature Cell Biology has now transitioned to a unified Rights Collection system which will allow our Author Services team to quickly and easily collect the rights and permissions required to publish your work. Approximately 10 days after your paper is formally accepted, you will receive an email in providing you with a link to complete the grant of rights. If your paper is eligible for Open Access, our Author Services team will also be in touch regarding any additional information that may be required to arrange payment for your article.

Please note that *Nature Cell Biology* is a Transformative Journal (TJ). Authors may publish their research with us through the traditional subscription access route or make their paper immediately open access through payment of an article-processing charge (APC). Authors will not be required to make a final decision about access to their article until it has been accepted. Find out more about Transformative Journals

Please use the following link for uploading these materials:
[Redacted]

Best regards,

Nyx Hills
Staff
Nature Cell Biology

On behalf of

Daryl J.V. David, PhD

Senior Editor, Nature Cell Biology
Consulting Editor, Nature Communications
Nature Portfolio

Heidelberger Platz 3, 14197 Berlin, Germany
Email: daryl.david@nature.com
ORCID: <https://orcid.org/0000-0002-9253-4805>

Reviewer #1:

Remarks to the Author:

The authors have carefully addressed all points raised in the initial review. I have no further comments on the current manuscript.

Reviewer #2:

Remarks to the Author:

The authors have done a good job of addressing the concerns of this reviewer. The addition of an explanation for the technical difficulties encountered while attempting to address this reviewer's concerns and a reasonable explanation for these difficulties will be helpful for other groups (and even future researchers in corresponding authors' groups). The other edits and additional data added to this manuscript greatly improve the presented study. This reviewer supports publication of this manuscript in Nature Cell Biology.

Reviewer #3:

Remarks to the Author:

Meier et al have done a nice job revising their manuscript. The new version was a pleasure to read, and it provides a thorough study of both the in vitro and in vivo behaviors of the network of three important +TIP proteins, Kar9, Bim1, and Bik1. In particular, I note that the revised manuscript includes a more comprehensive discussion of the domain organization of the proteins and how these proteins compare to other proteins known to undergo liquid-liquid phase separation. I also appreciate the addition of new quantitative data establishing that fusion of Kar9 puncta in vivo is not rare, and that the rates of fusion and fission are accelerated in strains with mutations in the key protein-protein interaction sites within the network. My earlier concerns about hexandiol are at least partially

alleviated by the data showing that about 70% of hexandiol-treated cells retain discernable astral MTs. These observations show that the treatment did not massively disrupt the microtubule cytoskeleton. New data show that levels of all three +TIPs at the ends of astral microtubules in vivo are strongly reduced by hexanediol and more cautious language is used now in the corresponding paragraph (lines 341-343). Overall, the manuscript is excellent and, in my view, deserves publication.

Final Decision Letter:

Dear Dr. Barral,

I am writing on behalf of my colleague, Dr. Daryl David, who is out of the office.

I am pleased to inform you that your manuscript, "Multivalency ensures persistence of a +TIP-body at specialized microtubule ends", has now been accepted for publication in Nature Cell Biology.

Please note that *Nature Cell Biology* is a Transformative Journal (TJ). Authors may publish their research with us through the traditional subscription access route or make their paper immediately

open access through payment of an article-processing charge (APC). Authors will not be required to make a final decision about access to their article until it has been accepted. Find out more about Transformative Journals

If you have not already done so, we strongly recommend that you upload the step-by-step protocols used in this manuscript to the Protocol Exchange (www.nature.com/protocolexchange), an open online resource established by Nature Protocols that allows researchers to share their detailed experimental know-how. All uploaded protocols are made freely available, assigned DOIs for ease of citation and are fully searchable through nature.com. Protocols and Nature Portfolio journal papers in which they are used can be linked to one another, and this link is clearly and prominently visible in the online versions of both papers. Authors who performed the specific experiments can act as primary authors for the Protocol as they will be best placed to share the methodology details, but the Corresponding Author of the present research paper should be included as one of the authors. By uploading your Protocols to Protocol Exchange, you are enabling researchers to more readily reproduce or adapt the methodology you use, as well as increasing the visibility of your protocols and papers. You can also establish a dedicated page to collect your lab Protocols. Further information can be found at www.nature.com/protocolexchange/about

All the best,

Christina

==

Christina Kary, PhD
Chief Editor
Nature Cell Biology
1 New York Plaza

Tel: +44 (0) 207 843 4924

** Visit the Springer Nature Editorial and Publishing website at www.springernature.com/editorial-and-publishing-jobs for more information about our career opportunities. If you have any questions please click here.**